https://doi.org/10.1038/s42003-022-04284-x — OPEN
# *CARD11* mutation and *HBZ* expression induce lymphoproliferative disease and adult T-cell leukemia/lymphoma

Takuro Kameda [1], Kotaro Shide[1], Ayako Kamiunten[1], Yasunori Kogure [2], Daisuke Morishita[3], Junji Koya[2], Yuki Tahira[1], Keiichi Akizuki[1], Takako Yokomizo-Nakano[4], Sho Kubota[4], Kosuke Marutsuka[5], Masaaki Sekine[1], Tomonori Hidaka[1], Yoko Kubuki[1], Yuichi Kitai[6], Tadashi Matsuda[6], Akinori Yoda[7], Takayuki Ohshima [8], Midori Sugiyama[3], Goro Sashida [4], Keisuke Kataoka[2,9], Seishi Ogawa [7] & Kazuya Shimoda [1✉]

Adult T-cell leukemia/lymphoma (ATL) is caused by human T-cell leukemia virus type 1 (HTLV-1). In addition to *HTLV-1 bZIP factor* (*HBZ*), a leukemogenic antisense transcript of HTLV-1, abnormalities of genes involved in TCR-NF-κB signaling, such as *CARD11*, are detected in about 90% of patients. Utilizing mice expressing CD4+ T cell-specific CARD11(E626K) and/or CD4+ T cell-specific *HBZ*, namely CARD11(E626K)$^{CD4\text{-}Cre}$ mice, *HBZ* transgenic (Tg) mice, and CARD11(E626K)$^{CD4\text{-}Cre}$;*HBZ* Tg double transgenic mice, we clarify these genes' pathogenetic effects. CARD11(E626K)$^{CD4\text{-}Cre}$ and *HBZ* Tg mice exhibit lymphocytic invasion to many organs, including the lungs, and double transgenic mice develop lymphoproliferative disease and increase CD4+ T cells in vivo. CARD11(E626K) and *HBZ* cooperatively activate the non-canonical NF-κB pathway, IRF4 targets, BATF3/IRF4/HBZ transcriptional network, MYC targets, and E2F targets. Most KEGG and HALLMARK gene sets enriched in acute-type ATL are also enriched in double transgenic mice, indicating that these genes cooperatively contribute to ATL development.

[1] Division of Hematology, Diabetes, and Endocrinology, Department of Internal Medicine, Faculty of Medicine, University of Miyazaki, Miyazaki, Japan. [2] Division of Molecular Oncology, National Cancer Center Research Institute, Tokyo, Japan. [3] Chordia Therapeutics, Fujisawa, Kanagawa, Japan. [4] Laboratory of Transcriptional Regulation in Leukemogenesis, International Research Center for Medical Sciences (IRCMS), Kumamoto University, Kumamoto, Japan. [5] Department of Anatomic Pathology, Miyazaki Prefectural Miyazaki Hospital, Miyazaki, Japan. [6] Department of Immunology, Graduate School of Pharmaceutical Sciences, Hokkaido University, Sapporo, Hokkaido, Japan. [7] Department of Pathology and Tumor Biology, Kyoto University, Kyoto, Japan. [8] Faculty of Pharmaceutical Sciences at Kagawa Campus, Tokushima Bunri University, Kagawa, Japan. [9] Division of Hematology, Department of Medicine, Keio University School of Medicine, Tokyo, Japan. ✉email: kshimoda@med.miyazaki-u.ac.jp

Adult T-cell leukemia/lymphoma (ATL) is an aggressive peripheral T-cell lymphoma that develops in about 5% of people infected with human T-cell leukemia virus type 1 (HTLV-1), usually after decades of clinical latency[1–5]. Viral oncogenes encoded by HTLV-1, namely *tax* and *HTLV-1 bZIP factor* (*HBZ*), are thought to play a major role in ATL development[6,7]. Tax possesses transforming activity in vitro, and Tax transgenic (Tg) mice develop cancer in vivo[8,9]. However, Tax is highly immunogenic, and its expression is suppressed and disrupted in many ATL cells[10,11]. In contrast, *HBZ*, which is encoded in the minus strand of the HTLV-1 genome, is transcribed in all ATL cases, and HBZ promotes ATL cell proliferation in vivo and induces T-cell lymphoma and inflammation in vivo[12–15].

The development of ATL involves not only viral oncogenes, but also presumably the accumulation of gene mutations in T cells during the long latency period from HTLV-1 infection in infancy to ATL onset later in life[4]. We previously reported that about 90% of ATL patients possessed mutations in genes involved in T-cell receptor (TCR)–NF-κB signaling and downstream or associated pathways[16]. Among them, mutations in *CARD11*, a cytoplasmic scaffolding protein required for antigen receptor-induced NF-κB activation in both T and B cells, were detected in 24% of ATL patients[16]. The most frequently observed CARD11 mutation, comprising 17% of the total, was the E626K gain-of-function mutation[16]. Gain-of-function mutations in *CARD11* were also reported in 9.6% of patients with diffuse large B-cell lymphoma (DLBCL) of the activated B-cell (ABC) subtype, and mice harboring active *CARD11* mutations in B cells developed aggressive B-cell lymphoproliferation, leading to early postnatal lethality[17–19].

Here we generated a mouse model involving the conditional expression of a CARD11(E626K) gain-of-function mutant, which is the most common mutation observed in ATL patients[16], and demonstrated that CARD11 activation in CD4+ T cells induced lymphadenopathy and T-cell infiltration of many organs. We also showed that the combination of HBZ expression and CARD11 mutation cooperatively accelerated the onset of lymphoproliferative disease, affected the expression of many genes, and served as the molecular basis of acute-type ATL.

## Results

**Generation of CARD11(E626K)$^{CD4\text{-}Cre}$ and *HBZ* Tg mice.** To investigate constitutively active CARD11 signaling in CD4+ T cells in vivo, we generated CARD11(E626K)$^{CD4\text{-}Cre}$ mice; this mutation is the homologue of human *CARD11*(E626K), the most commonly observed mutation in ATL patients (Supplementary Fig. 1a). Compared to WT CD4+ T cells, CD4+ T cells from CARD11(E626K)$^{CD4\text{-}Cre}$ mice demonstrated a 1.2-fold increase in the ratio of phosphorylated CARD11 Ser652 to CARD11 (Supplementary Fig. 2a). *HBZ* Tg mice were generated by *HBZ* cDNA expression under a mouse *CD4* promoter (Supplementary Fig. 1b). Crossing CARD11(E626K)$^{CD4\text{-}Cre}$ and *HBZ* Tg mice resulted in CARD11(E626K)$^{CD4\text{-}Cre}$;*HBZ* Tg mice.

Western blotting showed that compared with WT mice, the level of CARD11 in CD4+ T cells was increased by 3-fold and 2.5-fold in CARD11(E626K)$^{CD4\text{-}Cre}$ mice and CARD11(E626K)$^{CD4\text{-}Cre}$;*HBZ* Tg mice, respectively (Supplementary Fig. 2b). There was no difference between these mouse types regarding the CARD11 protein level. HBZ was detected in CD4+ T cells from *HBZ* Tg mice, but not in those from WT mice (Supplementary Fig. 2b). The protein level of HBZ in CARD11(E626K)$^{CD4\text{-}Cre}$;*HBZ* Tg mice was comparable to that in *HBZ* Tg mice.

The median survival time of CARD11(E626K)$^{CD4\text{-}Cre}$, *HBZ* Tg, and CARD11(E626K)$^{CD4\text{-}Cre}$;*HBZ* Tg mice was 14, 8, and 6 months, respectively (Fig. 1a).

**Lymphocyte invasion in many organs in CARD11(E626K)$^{CD4\text{-}Cre}$ and *HBZ* Tg mice.** In CARD11(E626K)$^{CD4\text{-}Cre}$ mice, leukocytosis and an increased number of CD4+ cells were observed in peripheral blood (Fig. 1b). Morphologically abnormal/atypical cells were rarely observed (Fig. 1c). CARD11(E626K)$^{CD4\text{-}Cre}$ mice did not show an increase in splenic nucleated cells (Fig. 2a). Normal murine splenic histological architecture contains clearly bordered white and red pulp. White pulp consists of aggregates of nucleated cells, mainly lymphocytes, while red pulp consists of a vascularized sinus meshwork. At 6 months after birth, the splenic histological architecture in CARD11(E626K)$^{CD4\text{-}Cre}$ mice was almost intact, but the interface between white and red pulp was uneven and obscure (Fig. 2b). This was attributed to focal infiltration of CD3+ T cells from the white pulp periarteriolar region into the red pulp. At 12 months after birth, areas of white pulp were decreased in number and size, while those of red pulp were larger, and the interface had become disrupted. CD3+ T cells had diffusely infiltrated both the shrunken white pulp and the expanding red pulp. The size and cellularity of the thymus was unchanged in CARD11(E626K)$^{CD4\text{-}Cre}$ mice (Supplementary Fig. 3).

Most CARD11(E626K)$^{CD4\text{-}Cre}$ mice developed lymphadenopathy. However, the swollen lymph nodes (LNs) were not big, with a mean diameter of about 3 mm. Lymphadenopathy was observed in 12 of 18 mice (67%) at 6 months after birth, and in 12 of 14 mice (86%) at 12 months after birth (Fig. 3a). The LN histological structure in WT mice consists of the cortex, paracortex, and medulla. Lymph follicles are located within the cortex, and the paracortex contains veins, with a high proportion of endothelial cells, into which circulating lymphocytes enter. At 6 months after birth, CARD11(E626K)$^{CD4\text{-}Cre}$ mice exhibited slightly shrunken follicles, a thickened cortex, and an expanded paracortex with CD3+ T-cell infiltration (Fig. 3b). The interface between the cortex and paracortex was obscured by infiltrating T cells. At 12 months, the normal LN architecture was barely present in CARD11(E626K)$^{CD4\text{-}Cre}$ mice. The cortex had become thickened and the follicles had disappeared. The paracortex was expanded by CD3+ T cells, and the interface was disrupted.

The most affected organ in CARD11(E626K)$^{CD4\text{-}Cre}$ mice was the lung. In WT mice, alveolar ducts, alveolar sacs, and alveoli have very thin walls invested with fine close-meshed networks of large thin-walled capillaries. In CARD11(E626K)$^{CD4\text{-}Cre}$ mice at 6 months after birth, focal lymphocyte infiltration was observed; both T and B cells had infiltrated the perivascular interstitium and occasionally the alveolar septa. Over time this infiltration increasingly consisted of CD3+ T cells, which formed diffuse lung lesions with thickening of the interstitium and alveolar septa at 12 months (Fig. 4).

*HBZ* Tg mice exhibited leukopenia and a decreased number of CD4+ T cells in peripheral blood (Fig. 1b). Morphologically abnormal/atypical cells were rarely observed (Fig. 1c). The number of nucleated cells in the spleen was comparable between *HBZ* Tg and WT mice (Fig. 2a). Histological examination at 6 months after birth revealed that the spleen was almost normal; however, at 12 months after birth, the white pulp exhibited expansion (Fig. 2b). CD3+ T cells had caused expansion of both white and red pulp, which disrupted their interface.

The thymus was shrunken in size in *HBZ* Tg mice at 6 months after birth. In one lobe of the thymus, the nucleated cell number in *HBZ* Tg mice was comparable to that in WT mice, and the number of CD4+ T cells was lower than that in WT mice (Supplementary Fig. 3).

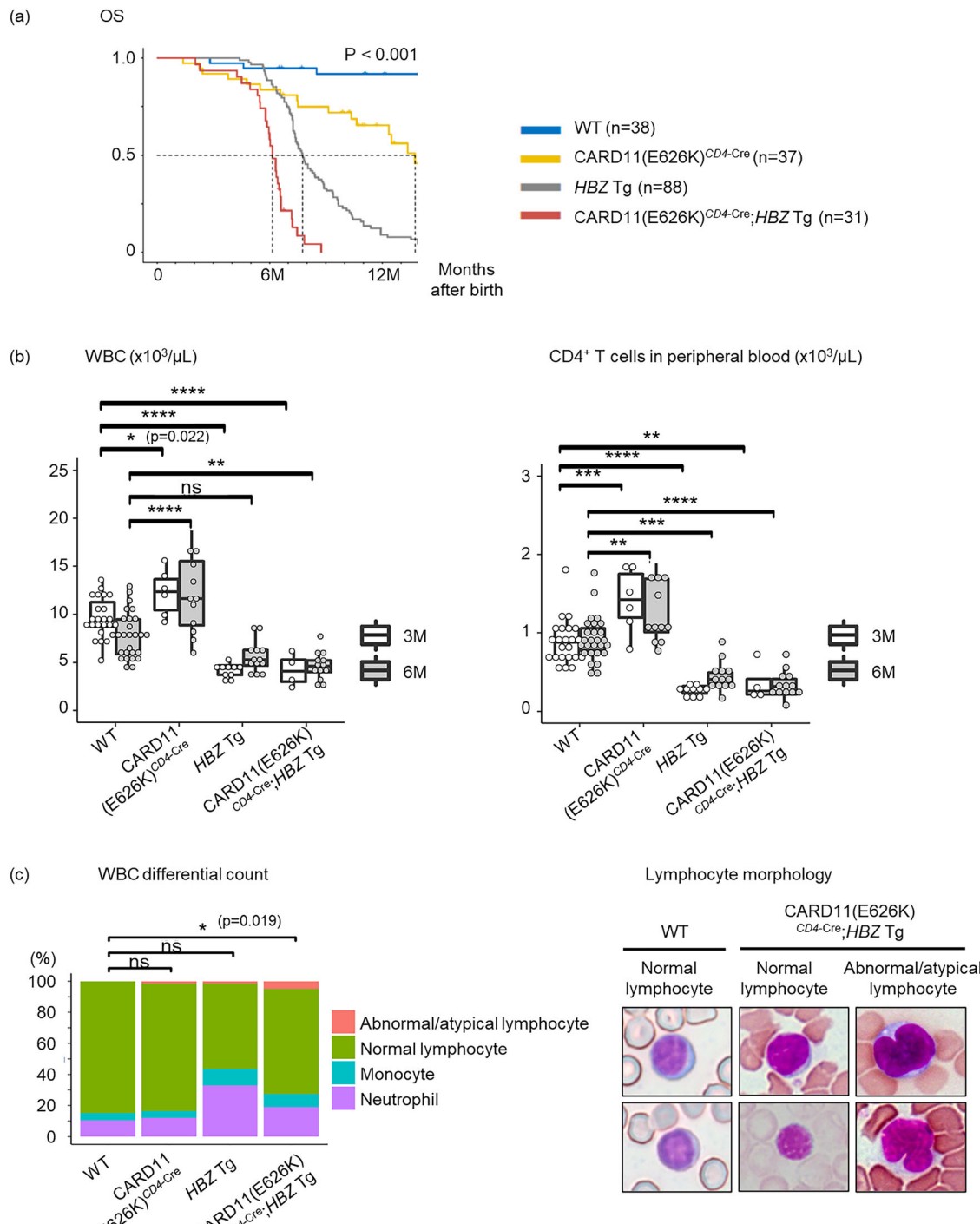

**Fig. 1 Overall survival and cells in peripheral blood and spleen in WT, CARD11(E626K)$^{CD4-Cre}$, *HBZ* Tg, and CARD11(E626K)$^{CD4-Cre}$;*HBZ* Tg mice.**
**a** Kaplan–Meier survival curves of mutant mice (wild type (WT), $n = 38$; CARD11(E626K)$^{CD4-Cre}$, $n = 37$; *HBZ* Tg, $n = 88$; CARD11(E626K)$^{CD4-Cre}$;*HBZ* Tg, $n = 31$). The median survival times of WT, CARD11(E626K)$^{CD4-Cre}$, *HBZ* Tg, and CARD11(E626K)$^{CD4-Cre}$;*HBZ* Tg mice, respectively, are as follows: not reached, 14, 8, and 6 months after birth. **b** The number of white blood cells and CD4$^+$ T cells in peripheral blood in WT ($n = 23$), CARD11(E626K)$^{CD4-Cre}$ ($n = 6$), HBZ Tg ($n = 10$), and CARD11(E626K)$^{CD4-Cre}$;*HBZ* Tg ($n = 4$) mice at 3 months after birth, and those in WT ($n = 27$), CARD11(E626K)$^{CD4-Cre}$ ($n = 12$), *HBZ* Tg ($n = 13$), and CARD11(E626K)$^{CD4-Cre}$;*HBZ* Tg ($n = 13$) mice at 6 months after birth. **c** The differential white blood cell count was performed manually, and a peripheral blood slide revealed that about 5% of lymphocytes in CARD11(E626K)$^{CD4-Cre}$;*HBZ* Tg mice demonstrated pleomorphic nuclear features (WT, $n = 5$; CARD11(E626K)$^{CD4-Cre}$, $n = 3$; *HBZ* Tg, $n = 3$; CARD11(E626K)$^{CD4-Cre}$;*HBZ* Tg, $n = 3$). The blood smear was stained with Wright-Giemsa. p values were calculated by the Tukey test after one-way ANOVA, and *, **, ***, and **** represent p values less than 0.05, 0.01, 0.001, and 0.0001, respectively.

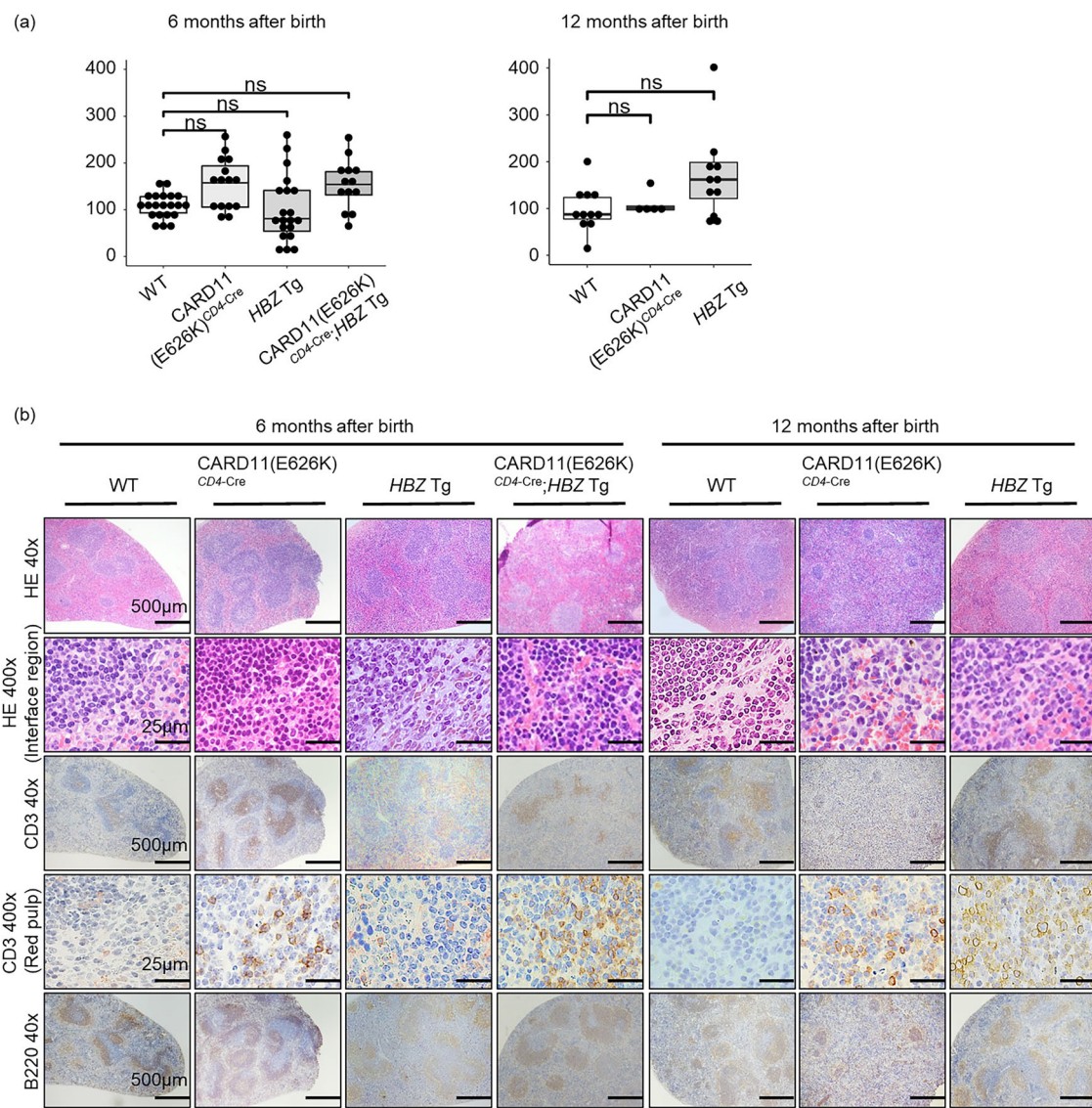

**Fig. 2 Number of nucleated cells and pathological findings of spleens in WT, CARD11(E626K)$^{CD4\text{-}Cre}$, *HBZ* Tg, and CARD11(E626K)$^{CD4\text{-}Cre}$;*HBZ* Tg mice. a** Absolute numbers of nucleated cells in the spleen at 6 months after birth (wild type (WT), $n = 21$; CARD11(E626K)$^{CD4\text{-}Cre}$, $n = 15$; *HBZ* Tg, $n = 20$; CARD11(E626K)$^{CD4\text{-}Cre}$;*HBZ* Tg, $n = 13$) and 12 months after birth (WT, $n = 11$; CARD11(E626K)$^{CD4\text{-}Cre}$, $n = 5$; *HBZ* Tg, $n = 12$). **b** Histologic analysis of the spleen at 6 and 12 months after birth ($n = 5$ for each mouse type). Spleens were stained with hematoxylin and eosin (HE) and anti-CD3 and anti-B220 antibodies. The interface between white and red pulp is shown at high magnification (HE 400×). CD3$^+$ T cells in red pulp are shown at high magnification (CD3 400×).

Lymphadenopathy was obvious at 12 months after birth in *HBZ* Tg mice (Fig. 3a). The frequency of lymphadenopathy at 12 months after birth was 67%, which was similar to that in a previous report[15]. The normal LN architecture was destroyed. The follicles were shrunken, the cortex was thickened, and the paracortex was expanded (Fig. 3b). CD3$^+$ T cells were mainly observed in the paracortex, and they infiltrated the cortex.

As in CARD11(E626K)$^{CD4\text{-}Cre}$ mice, the most affected organ in *HBZ* Tg mice was the lung. Diffuse infiltration of lymphocytes was observed at 6 months, and lymphocytes comprising mainly T cells invaded the perivascular interstitium and alveolar septa. The alveolar space was filled with exudate at 6 months after birth (Fig. 4). These abnormalities were much more pronounced at 12 months after birth. Severe lung findings were observed, with exudate and thickening of the interstitium and alveolar septa. These features, except for the exudate, resembled those in CARD11(E626K)$^{CD4\text{-}Cre}$ mice.

**The combination of *CARD11* mutation and *HBZ* expression induced the development of lymphoproliferative disease.** *HBZ* is thought to play a major role in ATL development[6,7] and is detected in CD4$^+$ T cells from almost all patients with acute-type ATL, while *CARD* mutations are found in about one-fourth of patients with ATL. We therefore analyzed how HBZ expression and CARD mutation affected the development of lymphoproliferative neoplasms in vivo. The subsequent analysis was performed at 6 months after birth since most CARD11(E626K)$^{CD4\text{-}Cre}$;*HBZ* Tg mice died within 9 months after birth. CARD11(E626K)$^{CD4\text{-}Cre}$;*HBZ* Tg mice exhibited leukopenia and a decreased number of CD4$^+$ T cells in peripheral blood (Fig. 1b). About 5% of lymphocytes demonstrated pleomorphic nuclear features, whereas other lymphocytes had nearly normal morphology (Fig. 1c).

In the spleen, neither splenomegaly nor an increased number of nucleated cells was observed (Fig. 2a, b). The splenic

(a)

| Age | Genotype | Lymphoadenopathy, n (%) | | p (vs. WT) |
| --- | --- | --- | --- | --- |
| | | Present | Absent | |
| 6 months | WT | 0 (0) | 21 (100) | — |
| | CARD11(E626K)$^{CD4\text{-}Cre}$ | 12 (67) | 6 (33) | <0.001 |
| | *HBZ* Tg | 5 (19) | 22 (81) | 0.118 |
| | CARD11(E626K)$^{CD4\text{-}Cre}$;*HBZ* Tg | 17 (85) | 3 (15) | <0.001 |
| 12 months | WT | 2 (11) | 16 (89) | — |
| | CARD11(E626K)$^{CD4\text{-}Cre}$ | 12 (86) | 2 (14) | <0.001 |
| | *HBZ* Tg | 8 (67) | 4 (33) | 0.008 |

(b)

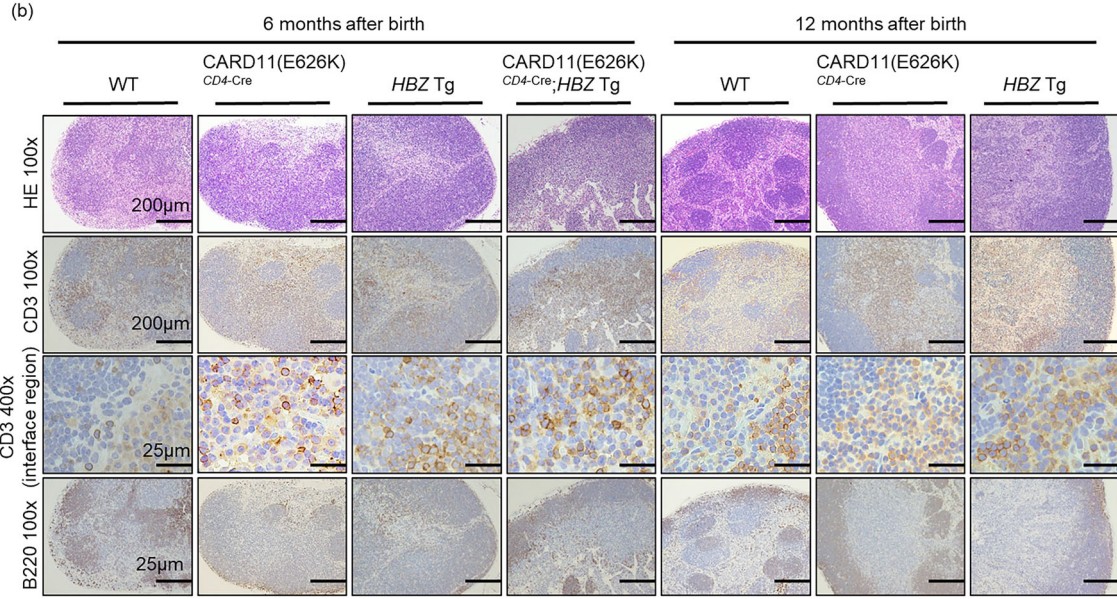

**Fig. 3 Lymphadenopathy and disruption of lymph node architecture in mutant mice. a** The frequency of lymphadenopathy at 6 and 12 months after birth. Swollen lymph nodes (LNs) are not large, with a mean diameter of about 3 mm. Differences between frequencies of lymphadenopathy were assessed by Fisher's exact test with Benjamini–Hochberg correction. **b** Histologic analysis of LNs at 6 and 12 months after birth (*n* = 5 for each mouse type). LNs were stained with hematoxylin and eosin (HE), and anti-CD3 and anti-B220 antibodies. The interface between the cortex and paracortex is shown with high magnification (CD3 400×).

architecture was destroyed and the interface between white and red pulp was disrupted. CD3$^+$ T cells were diffusely present throughout the spleen. The thymus was shrunken in size in CARD11(E626K)$^{CD4\text{-}Cre}$;*HBZ* Tg mice at 6 months after birth compared with WT mice (Supplementary Fig. 3). Accordingly, the numbers of nucleated cells and CD4$^+$CD8$^-$ T cells in one lobe of the thymus were lower in CARD11(E626K)$^{CD4\text{-}Cre}$;*HBZ* Tg mice compared with WT mice.

Lymphadenopathy was detected in 17 of 20 (85%) CARD11(E626K)$^{CD4\text{-}Cre}$;*HBZ* Tg mice (Fig. 3). The normal LN architecture was destroyed; follicles had almost disappeared, the cortex was thickened, and the paracortex was expanded. CD3$^+$ T cells were observed mainly in the paracortex and had massively infiltrated the cortex, leading to the highly disrupted interface.

The most affected organ in CARD11(E626K)$^{CD4\text{-}Cre}$;*HBZ* Tg mice was the lung. The lung interstitium and alveolar septa were thickened with infiltrating cells consisting mainly of lymphocytes (Fig. 4). CD3$^+$ T cells had accumulated around capillary blood vessels. The alveolar space had filled with exudate, and subsequently with macrophages, and consequently almost no air spaces were observed. Mice with both CARD11 mutation and *HBZ* expression developed the aggressive lymphoproliferative disease.

**Increment of CD4$^+$ T cells, effector/memory T cells (Tem), and regulatory T cells (Treg) in CARD11(E626K)$^{CD4\text{-}Cre}$;*HBZ* Tg mice.** Since ATL is a neoplasm of CD4$^+$ T lymphocytes, and both CARD11(E626K) and *HBZ* were expressed in CD4$^+$ T cells in our mice, we next evaluated the absolute numbers of CD4$^+$ T cells and their subsets, namely CD4$^+$CD44$^+$CD62L$^-$ Tem and CD4$^+$CD25$^+$ Treg, together with CD8$^+$ T cells, in the body of each mouse as the sum of their numbers in the whole bone marrow and whole spleen[20,21] (Fig. 5, Supplementary Fig. 4). In CARD11(E626K)$^{CD4\text{-}Cre}$ mice, the numbers of CD4$^+$CD8$^-$ T cells, Tem, Treg, and CD4$^-$CD8$^+$ T cells per mouse were not increased compared with WT mice. In *HBZ* Tg mice, there was a decrease in the number of CD4$^-$CD8$^+$ T cells per mouse at 6 months after birth, and an increase of CD4$^+$CD8$^-$ T cells and Tem per mouse at 12 months after birth. HBZ expression had no effect on the number of Treg. In CARD11(E626K)$^{CD4\text{-}Cre}$;*HBZ* Tg mice, the numbers of CD4$^+$CD8$^-$ T cells, Tem, and Treg were increased compared with WT mice. Consistent with this, the numbers of CD44$^+$ and FOXP3$^+$ cells were increased in LNs and lung in CARD11(E626K)$^{CD4\text{-}Cre}$;*HBZ* Tg mice, and the ratio of Ki-67 positive cells among Tem and Treg in LNs were increased compared with WT mice (Supplementary Figs. 5, 6); this indicated a cooperative positive effect of *CARD11* mutation and *HBZ* expression on the proliferation of each cell type. The number of

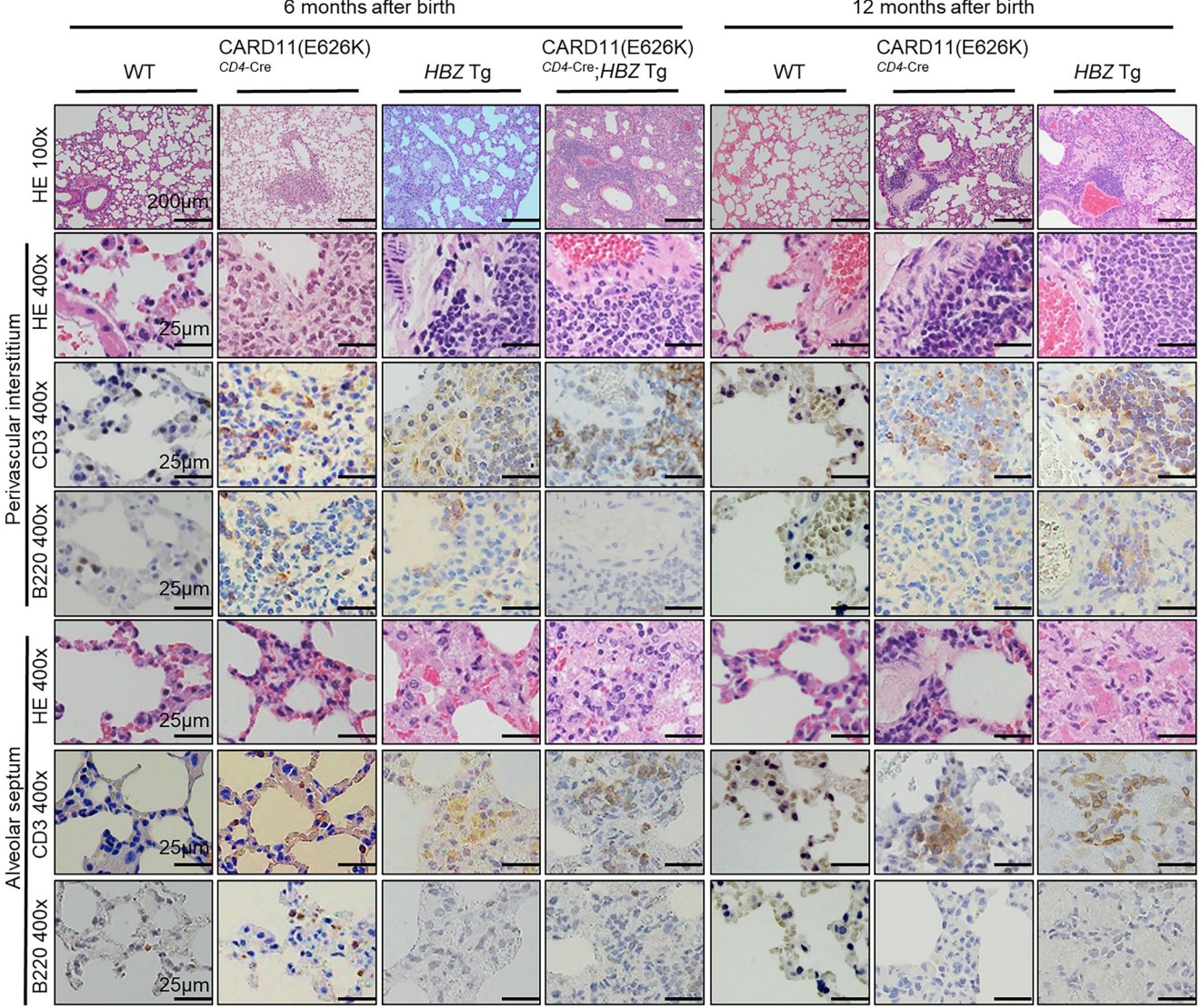

**Fig. 4 Lymphocyte infiltration into the lung in mutant mice.** Histologic analysis of the lung at 6 and 12 months after birth ($n = 5$ for each mouse type). Lungs were stained with hematoxylin and eosin (HE), as well as anti-CD3 and anti-B220 antibodies. Lymphocyte infiltration into the lung perivascular interstitium and alveolar septa is observed in mutant mice at 6 and 12 months.

CD4−CD8+ T cells in CARD11(E626K)$^{CD4\text{-}Cre}$;*HBZ* Tg mice was comparable to that in WT mice.

A hallmark of malignant transformation is the ability of a transplanted tumor to proliferate in another host. We transplanted splenic CD4+ T cells from WT, CARD11(E626K)$^{CD4\text{-}Cre}$, *HBZ* Tg, or CARD11(E626K)$^{CD4\text{-}Cre}$;*HBZ* Tg mice into immunocompromised NOD/Shi-scid/IL2rγnull (NOG) mice. At 18 weeks after transplantation, no overt tumors were visible in recipient mice. In recipient mice transplanted with CD4+ T cells from CARD11(E626K)$^{CD4\text{-}Cre}$;*HBZ* Tg mice, CD3+ T cells were detected in the spleen, whereas CD3+ T cells were rare in the spleens of recipient mice transplanted with CD4+ T cells from WT, CARD11(E626K)$^{CD4\text{-}Cre}$, or *HBZ* Tg mice (Supplementary Fig. 7). Some small LNs were detected in recipient mice transplanted with CD4+ T cells from CARD11(E626K)$^{CD4\text{-}Cre}$;*HBZ* Tg mice. These LNs contained mostly CD3+ T cells, and about 20% of the CD3+ T cells expressed Ki-67, suggesting that CARD11(E626K)$^{CD4\text{-}Cre}$;*HBZ* Tg mice developed lymphoproliferative disease. On the other hand, the absence of CD3+ T cells in the spleen and the absence of LNs containing CD3+ T cells in the recipient mice transplanted with CARD11(E626K)

$^{CD4\text{-}Cre}$ and *HBZ* Tg CD4+ T cells indicate that they developed inflammation, but not neoplasm.

**Gene expression changes due to the CARD11 mutant and *HBZ* expression.** To understand the basis for the phenotypes induced by the CARD11 mutant and *HBZ* expression in vivo, we performed gene expression profiling of Tem derived from WT, CARD11(E626K)$^{CD4\text{-}Cre}$, *HBZ* Tg, and CARD11(E626K)$^{CD4\text{-}Cre}$;*HBZ* Tg mice. Unsupervised hierarchical clustering of the global gene expression signatures revealed three major branches in the gene expression hierarchy (Supplementary Fig. 8a). WT and CARD11(E626K)$^{CD4\text{-}Cre}$ samples comprised the first branch, *HBZ* Tg samples comprised the second, and CARD11(E626K)$^{CD4\text{-}Cre}$;*HBZ* Tg samples comprised the third. The principal component analysis also showed that the CARD11 mutant and *HBZ* expression induced distinct transcriptomes (Supplementary Fig. 8b).

Compared with WT mice, the CARD11(E626K)$^{CD4\text{-}Cre}$, *HBZ* Tg, and CARD11(E626K)$^{CD4\text{-}Cre}$;*HBZ* Tg mice showed upregulation of 17, 877, and 508 genes, respectively, and downregulation

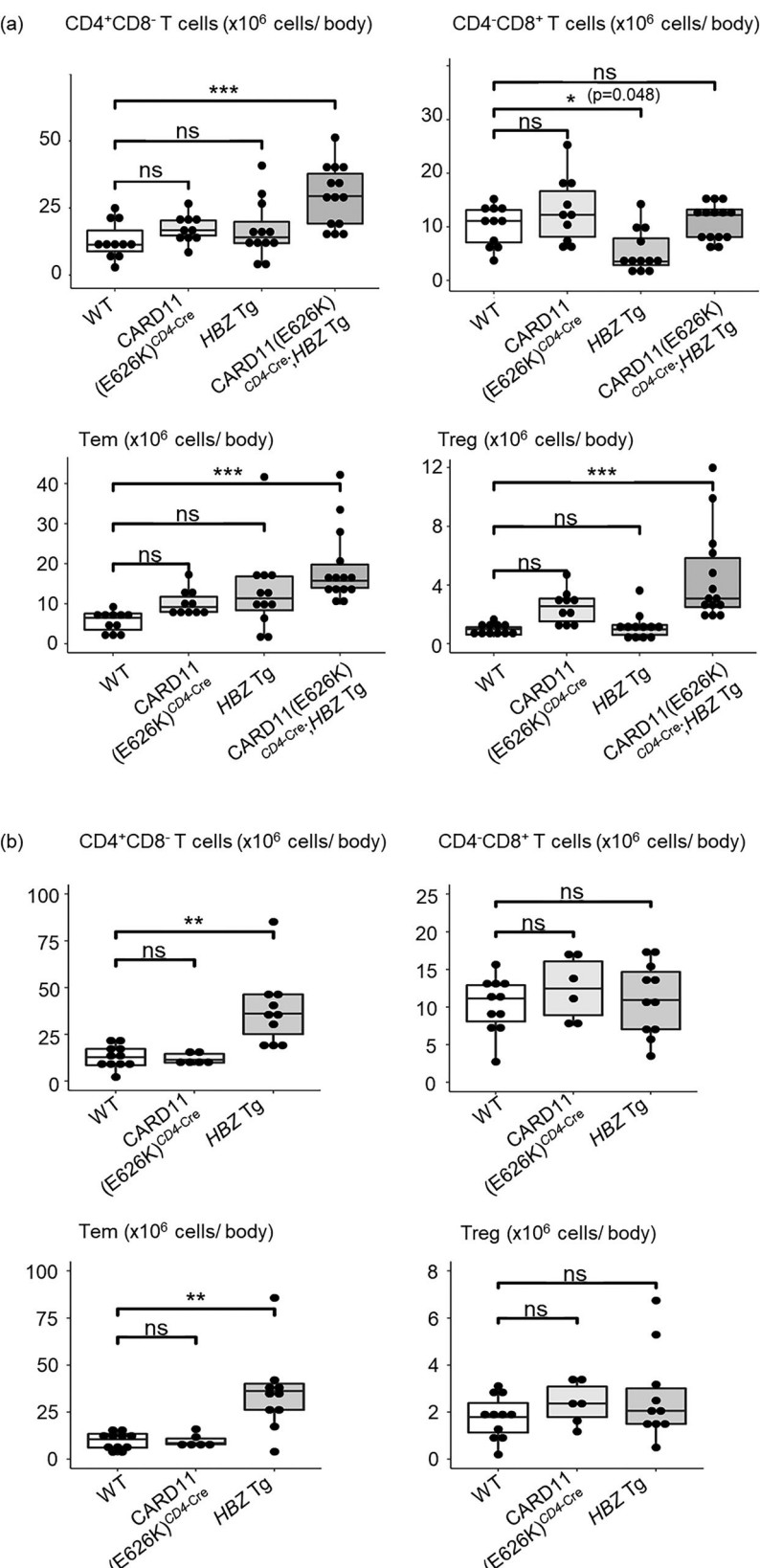

**Fig. 5 Increased numbers of CD4+ T cells and their subsets per body in mutant mice.** Absolute numbers of CD4+CD8− T cells, CD4-CD8+ T cells, CD4+CD44+CD62L− effector/memory T cells (Tem), and CD4+CD25+ regulatory T cells (Treg) per mouse in wild type (WT) ($n = 11$), CARD11(E626K)$^{CD4}$-Cre ($n = 10$), *HBZ* Tg ($n = 12$), and CARD11(E626K)$^{CD4-Cre}$;*HBZ* Tg ($n = 14$) mice at 6 months after birth (**a**), and those in WT ($n = 11$), CARD11(E626K)$^{CD4}$-Cre ($n = 6$), and *HBZ* Tg ($n = 11$) mice at 12 months after birth (**b**). *p* values were calculated by the Tukey test after one-way ANOVA, and *, **, and *** represent *p* values less than 0.05, 0.01, and 0.001, respectively.

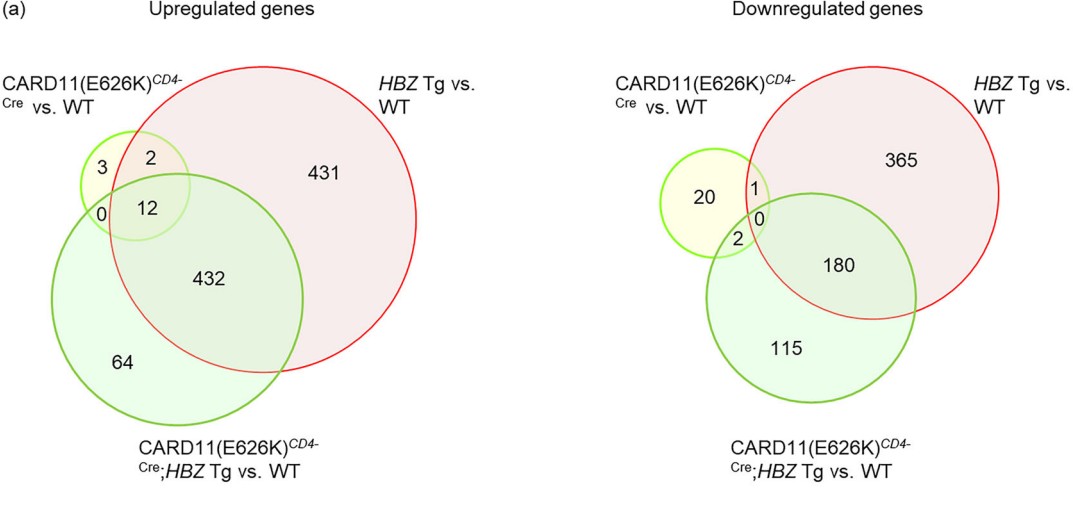

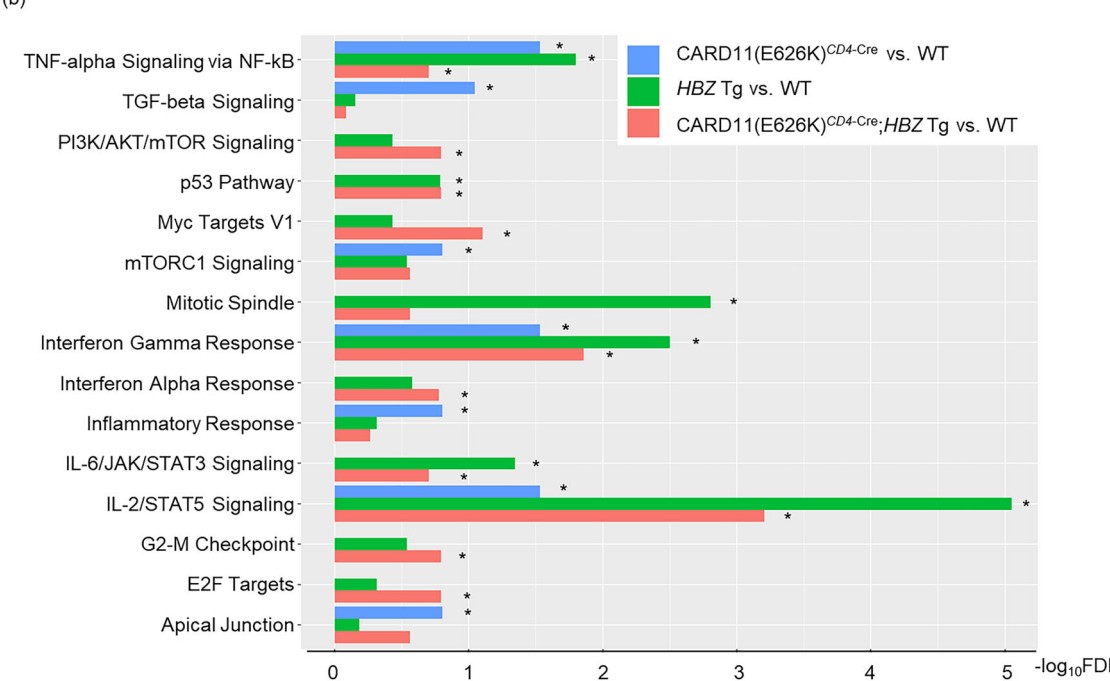

**Fig. 6 Uniquely or commonly activated pathways in CARD11(E626K)$^{CD4\text{-Cre}}$, *HBZ* Tg, and CARD11(E626K)$^{CD4\text{-Cre}}$;*HBZ* Tg mice. a** Venn diagrams of the overlap between significantly up- and downregulated genes in CARD11(E626K)$^{CD4\text{-Cre}}$ ($n = 3$) vs. WT ($n = 3$) mice, *HBZ*-Tg ($n = 4$) vs. WT mice, and CARD11(E626K)$^{CD4\text{-Cre}}$;*HBZ* Tg ($n = 3$) vs. WT mice. Splenic effector/memory T cells (Tem) from 4–6-month-old mice were sorted and underwent expressional analysis. Differentially expressed genes between sample groups were identified, using cut-offs of fold change (FC) > 1.2 and an FDR of <0.1. **b** Bar plots of significantly (FDR < 0.25) upregulated KEGG pathways. Some pathways are uniquely activated in CARD11(E626K)$^{CD4\text{-Cre}}$ mice, *HBZ* Tg mice, or CARD11(E626K)$^{CD4\text{-Cre}}$;*HBZ* Tg mice. On the other hand, IL-2/STAT5 signaling and interferon-gamma response are commonly activated in all three types of mice.

of 23, 546, and 297 genes, respectively (Fig. 6a). KEGG pathway analysis showed that TGF-beta signaling, mTORC1 signaling, inflammatory response, and apical junction were uniquely activated in CARD11(E626K)$^{CD4\text{-Cre}}$ mice; mitotic spindle signaling was uniquely activated in *HBZ* Tg mice; and PI3K/AKT/mTOR signaling, MYC target V1, G2M checkpoint, E2F target, and interferon alpha response were uniquely activated in CARD11(E626K)$^{CD4\text{-Cre}}$;*HBZ* Tg mice (Fig. 6b). In contrast, TNF alpha signaling via NF-kappa B, IL-2/STAT5 signaling and interferon-gamma response were activated in CARD11(E626K)$^{CD4\text{-Cre}}$, *HBZ* Tg, and CARD11(E626K)$^{CD4\text{-Cre}}$;*HBZ* Tg mice. The

p53 pathway and IL-6/JAK/STAT3 signaling were activated in both *HBZ* Tg and CARD11(E626K)$^{CD4\text{-Cre}}$;*HBZ* Tg mice, but not in CARD11(E626K)$^{CD4\text{-Cre}}$ mice.

CARD activates the NF-κB signaling pathway. GSEA showed heavy enrichment of canonical NF-κB pathway gene sets[22] in CARD11(E626K)$^{CD4\text{-Cre}}$ mice, but not in *HBZ* Tg or CARD11(E626K)$^{CD4\text{-Cre}}$;*HBZ* Tg mice (Supplementary Fig. 9). Non-canonical NF-κB pathway gene sets[23,24] were significantly enriched in CARD11(E626K)$^{CD4\text{-Cre}}$;*HBZ* Tg mice. Consistent with these results, high nuclear levels of RelA (p65) and p50 were detected in splenic CD4$^+$ cells from CARD11(E626K)$^{CD4\text{-Cre}}$

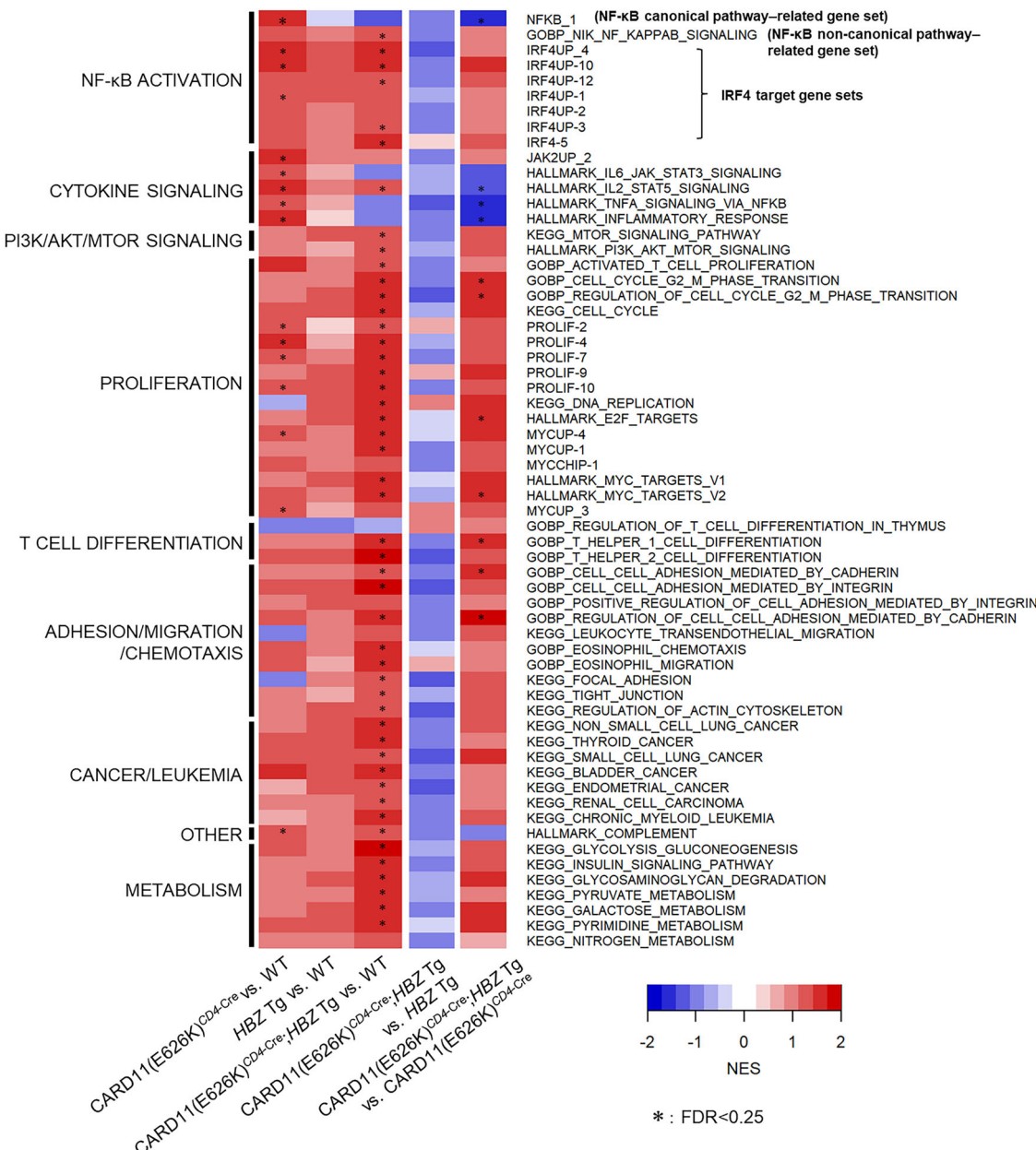

**Fig. 7 Heatmap overview of gene sets showing significant enrichment by GSEA in one or more mutant Tem compared to WT Tem.** Comparisons of results are shown in the following order: CARD11(E626K)$^{CD4-Cre}$ vs. WT, *HBZ* Tg vs. WT, CARD11(E626K)$^{CD4-Cre}$;*HBZ* Tg vs. WT, CARD11(E626K)$^{CD4-Cre}$;*HBZ* Tg vs. *HBZ* Tg, and CARD11(E626K)$^{CD4-Cre}$;*HBZ* Tg vs. CARD11(E626K)$^{CD4-Cre}$.

mice, indicating that the canonical NF-κB pathway was activated by *CARD11* mutation (Supplementary Fig. 10). In splenic CD4$^+$ cells from CARD11(E626K)$^{CD4-Cre}$;*HBZ* Tg mice, high nuclear levels of RelB were detected, indicating that the non-canonical NF-κB pathway was activated by the combined effects of *CARD11* mutation and *HBZ* expression.

*HBZ* mRNA and its transcript directly bind to the promoter regions of many genes. The selective apoptosis- and proliferation-related genes directly regulated by *HBZ* mRNA[25] were enriched in both *HBZ* Tg and CARD11(E626K)$^{CD4-Cre}$;*HBZ* Tg mice, and the enrichment of HBZ target gene sets[26], including indirect ones, were also observed in both mice (Supplementary Fig. 11).

The gene set enrichment pattern in Tem differed drastically between CARD11(E626K)$^{CD4-Cre}$ and *HBZ* Tg mice (Fig. 7). Gene sets related to cytokine signaling and inflammation were heavily enriched in CARD11(E626K)$^{CD4-Cre}$ mice. CARD

activation and *HBZ* expression might cooperatively affect global gene expression. In CARD11(E626K)$^{CD4-Cre}$;*HBZ* Tg mice, there was greater enrichment of gene sets related to the non-canonical NF-κB pathway, IRF4 targets, proliferation, T-cell differentiation, and cell adhesion than in WT mice. The gene sets whose expression was correlated with chromosomal instability were more enriched in CARD11(E626K)$^{CD4-Cre}$;*HBZ* Tg mice (Supplementary Fig. 12), suggesting that the combination of *HBZ* expression and *CARD11* mutation alter genes associated with genomic stability and contribute to the development of ATL[27].

Both *HBZ* Tg and CARD11(E626K)$^{CD4-Cre}$;*HBZ* Tg mice exhibited similar phenotypes, with the major difference being disease severity or timing. *HBZ* Tg mice at 6 months after birth exhibited a milder phenotype than CARD11(E626K)$^{CD4-Cre}$;*HBZ* Tg mice at 6 months after birth, but *HBZ* Tg mice at 12 months after birth exhibited similar severity as CARD11(E626K)$^{CD4-Cre}$;

*HBZ* Tg at 6 months after birth. We then compared the gene expression between *HBZ* Tg mice at 8–12 months and CARD11(E626K)$^{CD4\text{-Cre}}$;*HBZ* Tg mice at 4–6 months (Supplementary Fig. 13a). A total of 430 genes and 227 genes were commonly up- and downregulated, respectively, and target gene sets of HBZ were highly enriched in both *HBZ* Tg mice at 8–12 months and CARD11(E626K)$^{CD4\text{-Cre}}$;*HBZ* Tg mice at 4–6 months (Supplementary Figs. 11, 13b). Even though many genes were commonly up- or downregulated in both mouse types, there were genes that were uniquely up- or downregulated in only one type. Hierarchical clustering analysis also demonstrated that *HBZ* Tg mice at 8–12 months comprised a different gene expression hierarchy branch than CARD11(E626K)$^{CD4\text{-Cre}}$;*HBZ* Tg mice at 4–6 months (Supplementary Fig. 13c). The uniquely upregulated gene sets in CARD11(E626K)$^{CD4\text{-Cre}}$;*HBZ* Tg mice at 4–6 months ($n = 78$) were highly enriched in acute-type ATL patient samples compared with those from HTLV-1 carriers (Supplementary Fig. 13d). On the other hand, uniquely upregulated gene sets in *HBZ* Tg mice at 8–12 months ($n = 2970$) did not exhibit this enrichment. These observations suggested that CARD11(E626K)$^{CD4\text{-Cre}}$;*HBZ* Tg mice and *HBZ* Tg mice had some oncogenic signals in common, but they did not completely overlap.

Finally, we compared the enriched KEGG and HALLMARK gene sets between human samples obtained from patients with acute-type ATL[16] and samples from CARD11(E626K)$^{CD4\text{-Cre}}$, *HBZ* Tg, and CARD11(E626K)$^{CD4\text{-Cre}}$;*HBZ* Tg mice. As shown in Fig. 8a, about one-third of KEGG and HALLMARK gene sets were enriched in human acute-type ATL samples but not in healthy controls. About 10% of the enriched gene sets of human acute-type ATL samples from both the KEGG and HALLMARK databases were also enriched in CARD11(E626K)$^{CD4\text{-Cre}}$, and none of them was enriched in HBZ Tg mice. In contrast, CARD11(E626K)$^{CD4\text{-Cre}}$;*HBZ* Tg mice demonstrated enrichment of 74% and 94% of the enriched gene sets of human acute-type ATL samples from the two databases, respectively. The remaining gene sets that were only enriched in human acute-type ATL samples but not in CARD11(E626K)$^{CD4\text{-Cre}}$;*HBZ* Tg mice included many involved in metabolism (Supplementary Table 1). In addition, the NOTCH signaling pathway was only enriched in human acute-type ATL samples and not in Tem from CARD11(E626K)$^{CD4\text{-Cre}}$;*HBZ* Tg mice (Fig. 8b). Since there were many mutations besides those involving the TCR-NF-κB pathway in acute-type ATL patients, these other mutations might be responsible for the activation of the other signaling pathways that were enriched in human ATL samples but not in CARD11(E626K)$^{CD4\text{-Cre}}$;*HBZ* Tg mice.

The BATF3/IRF4/HBZ transcriptional network was reported to be essential for the proliferation of cells from an ATL cell line[28]. As shown in Fig. 9a, this network was enriched in both CARD11(E626K)$^{CD4\text{-Cre}}$;*HBZ* Tg mice and human acute-type ATL samples, but not in CARD11(E626K)$^{CD4\text{-Cre}}$ mice, *HBZ* Tg mice, or HTLV-1 carrier samples. *BATF3* was upregulated in both *HBZ* Tg and CARD11(E626K)$^{CD4\text{-Cre}}$;*HBZ* Tg mice (Fig. 9b). There was a 2.0-fold increase in IRF-4 protein levels in CD4$^+$ T cells from CARD11(E626K)$^{CD4\text{-Cre}}$;*HBZ* Tg mice compared with those from WT mice (Fig. 9c). Increased nuclear IRF4 protein was also observed in CD4$^+$ T cells from CARD11(E626K)$^{CD4\text{-Cre}}$;*HBZ* Tg mice, but not in those from *HBZ* Tg mice (Fig. 9d, e). In line with this, many IRF4 target gene sets were enriched in CARD11(E626K)$^{CD4\text{-Cre}}$;*HBZ* Tg mice but not in *HBZ* Tg mice (Supplementary Fig. 14). In CARD11(E626K)$^{CD4\text{-Cre}}$;*HBZ* Tg mice, the elevated BATF3 and IRF4 levels might induce the activation of the BATF3/IRF4/HBZ transcriptional network. Positive enrichment of proliferation-related gene sets such as MYC_TARGETS and E2F_TARGETS was also confirmed in CARD11(E626K)$^{CD4\text{-Cre}}$;*HBZ* Tg mice, highlighting the cooperative effect between CARD11(E626K)$^{CD4\text{-Cre}}$ and *HBZ* (Supplementary Fig. 15).

## Discussion

We showed here that the combination of constitutive *HBZ* expression and a gain-of-function mutation in CARD11 in CD4$^+$ T cells induced lethal lymphoproliferative disease in vivo, which includes LN swelling and lymphocyte invasion of many organs, especially the lungs. *HBZ* expression and *CARD11* mutation cooperatively increased the numbers of CD4$^+$ T cells, Tem, and Treg in vivo, and upregulated *BATF3* and IRF4, MYC target genes, and E2F target genes in CD4$^+$ T cells. These findings, and the fact that many of the KEGG and HALLMARK gene sets that were enriched in acute-type ATL patient samples were also enriched in CARD11(E626K)$^{CD4\text{-Cre}}$;*HBZ* Tg mice, indicate that the combination of *HBZ* expression and *CARD11* mutation forms the pathological basis for ATL development.

CARD11(E626K)$^{CD4\text{-Cre}}$ mice developed leukocytosis in peripheral blood, lymphadenopathy, and T-cell organ invasion, and exhibited shorter survival. The majority of infiltrating cells were Tem, which is consistent with previous reports that ATL clones are preserved in T memory stem cells (Tscm) and that the majority of ATL cells are conventional memory T cells[29]. However, the degree of leukocytosis was mild and the spleen was only minimally affected. This phenotype induced by the constitutive activation of CARD11 in T cells is very different from that in B cells. A mouse model that conditionally expressed the human DLBCL-derived CARD11(L225LI) mutant in B cells developed a rapidly lethal lymphoproliferative disorder with massive splenomegaly associated with a drastic increment in B-cell number[19]. Our observations might indicate that the in vivo signaling and other effects caused by constitutive activation of CARD11/BCL10/MALT1 (CBM) signaling differ between B and T cells, although the *CARD11* mutation is frequently observed in both B-cell malignancies, as the DLBCL ABC type, and in T-cell malignancies, as ATL. These ideas are consistent with the clinical features of the B-cell expansion with NF-κB and T-cell anergy (BENTA) syndrome, which is caused by germline gain-of-function mutations in *CARD11*[30]. Patients with BENTA syndrome exhibit splenomegaly and an increased number of B cells in PB, but a normal number of T cells[31,32]. The absence of abnormal growth of transplanted CD4$^+$ T cells with the *CARD11* mutation and the lack of tumor development in recipient mice provide confirmation that CARD11(E626K)$^{CD4\text{-Cre}}$ mice developed inflammation rather than neoplasia.

*HBZ* was found to be constitutively expressed in many ATL cases, and *HBZ* Tg mice were reported to develop lymphoma and systemic inflammation[15,33]. The phenotype of our *HBZ* Tg mice resembled that in previous reports[15,33]. Lymphadenopathy was observed in 66.7% of *HBZ* Tg mice at 12 months after birth. In a report by Satou, 37.8% of *HBZ* Tg mice developed T-cell lymphoma after 16 months[15]. In our *HBZ* Tg mice, the increment of CD4$^+$ T cells and Tem in the body was not observed at 6 months after birth, but became prominent at 12 months after birth. A long period of time might be required for the development of lymphoma in *HBZ* Tg mice.

Since HTLV-1-infected CD4$^+$ T cells transformed into ATL cells after developing gene mutations, we next examined the effect of the CARD11(E626K) mutation on *HBZ* Tg. CARD11(E626K)$^{CD4\text{-Cre}}$;*HBZ* Tg mice exhibited aggressive lymphoproliferative disease. The numbers of CD4$^+$ T cells, Tem, and Treg per mouse increased due to the cooperative effect of *CARD11* mutation and *HBZ* expression. There was increased infiltration of lymphocytes into the perivascular region and alveolar septum of the lung, causing almost complete obliteration of the alveolar space in

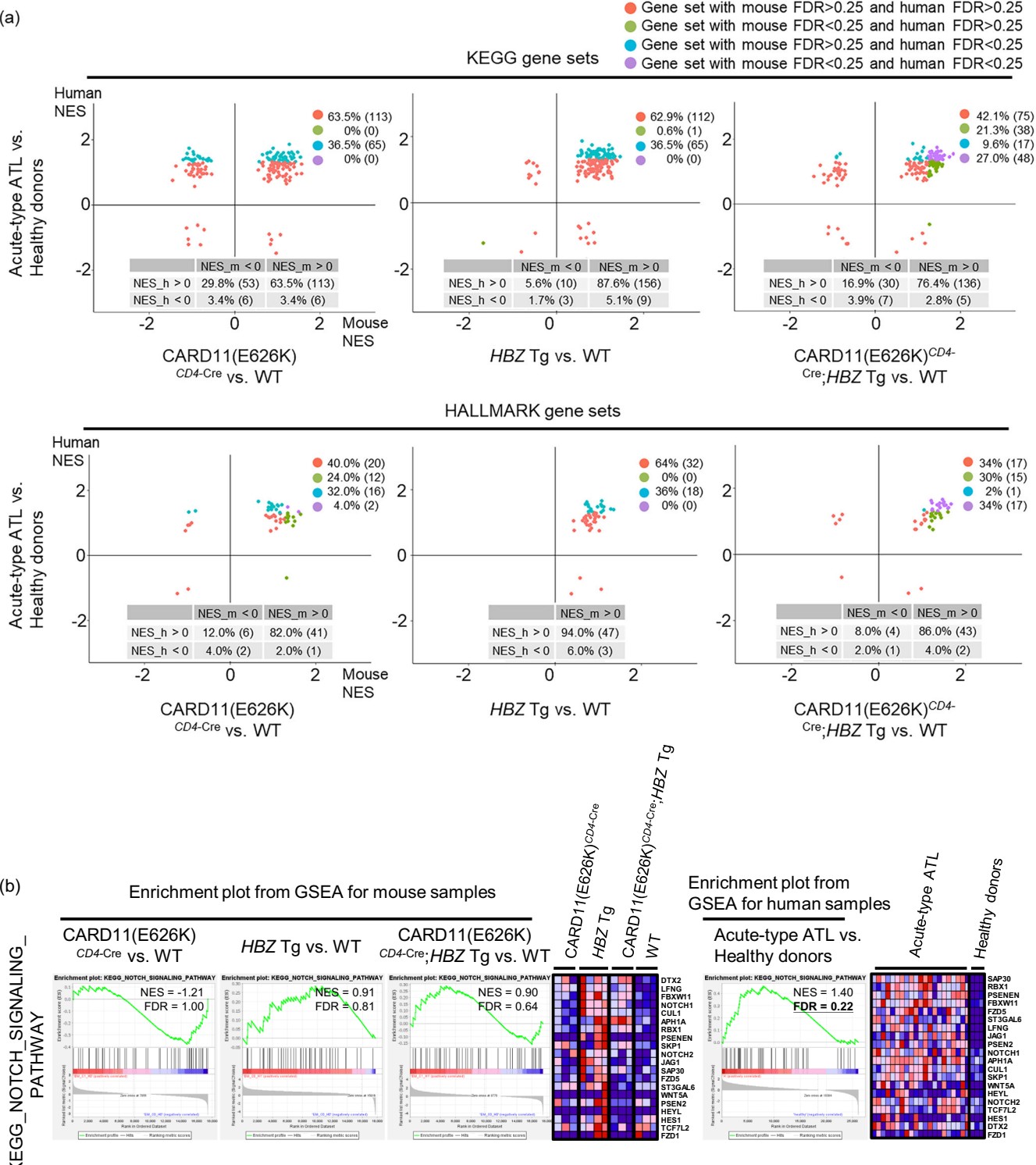

**Fig. 8 Gene expression signatures in CARD11(E626K)$^{CD4\text{-}Cre}$;*HBZ* Tg mice recapitulate many of those in human acute-type ATL samples. a** Normalized enrichment scores (NESs) in gene set enrichment analysis (GSEA) of the entire KEGG and HALLMARK gene sets show that mice and humans share many of the same enriched gene sets. Each gene set is indicated by a dot and is highlighted in a different color according to the respective false discovery rate (FDR) values for mice and humans, with a common cutoff value 0.25. Each plot legend indicates the frequency (number) of significant/non-significant gene sets, based on the common FDR cutoff values in both mice and humans. In the table at the bottom of each plot, the frequency (number) of pathways is indicated, based on the positive/negative values for both mouse NES (NES_m) and human NES (NES_h). **b** Notch signaling is enriched in the acute-type ATL samples, but not in CARD11(E626K)$^{CD4\text{-}Cre}$;*HBZ* Tg mice. Enrichment plots of the KEGG Notch signaling gene set in mice and humans are shown with their NESs and FDRs.

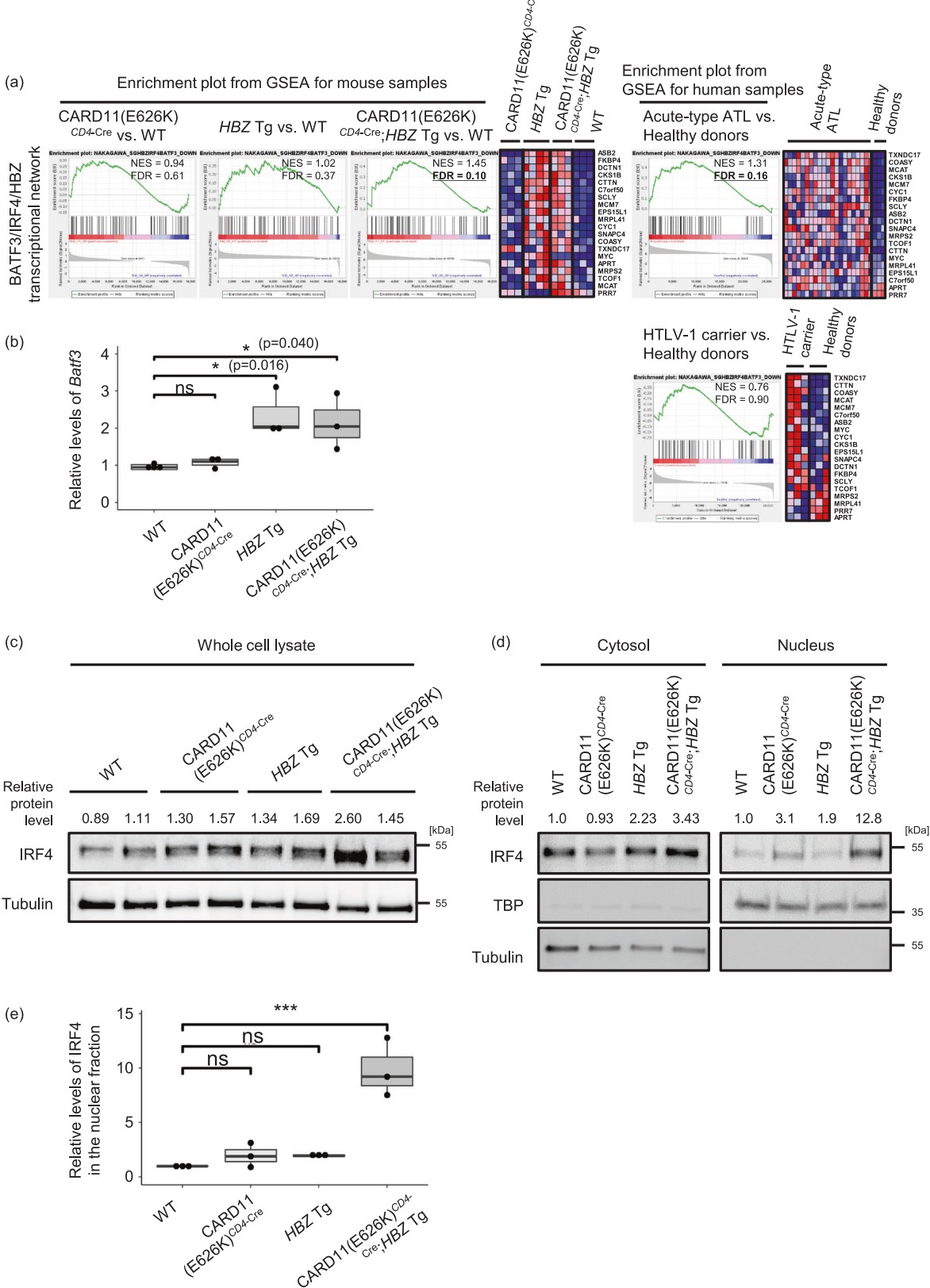

CARD11(E626K)$^{CD4\text{-}Cre}$;*HBZ* Tg mice. These mice demonstrated lymphadenopathy and complete destruction of the normal LN architecture. Further, T-cell invasion and growth in the spleen and LNs were observed in recipient mice transplanted with CD4$^+$ T cells from CARD11(E626K)$^{CD4\text{-}Cre}$;*HBZ* Tg mice. These observations support the theory that constitutive CARD11

activation and HBZ expression have a cooperative effect and cause the development of lymphoproliferative disease.

Gene set enrichment patterns differed drastically between CARD11(E626K)$^{CD4\text{-}Cre}$ and *HBZ* Tg mice. Canonical NF-κB pathway gene sets were enriched in CARD11(E626K)$^{CD4\text{-}Cre}$ mice. It was reported that activation of the canonical NF-κB

**Fig. 9 Formation of the BATF3/IRF4/HBZ transcriptional network in CARD11(E626K)$^{CD4\text{-}Cre}$;_HBZ_ Tg mice. a** CARD11(E626K)$^{CD4\text{-}Cre}$;_HBZ_ Tg mice recapitulated a transcriptional network change that is characteristic of ATL. The BATF3/IRF4/HBZ transcriptional network[28] is enriched in both CARD11(E626K)$^{CD4\text{-}Cre}$;_HBZ_ Tg mice and human acute-type ATL samples. Enrichment plots of each gene set in mice and humans are shown with their NESs and FDRs. **b** Quantitative PCR for _Batf3_ (WT, $n = 4$; CARD11(E626K)$^{CD4\text{-}Cre}$, $n = 3$; _HBZ_ Tg, $n = 3$; CARD11(E626K)$^{CD4\text{-}Cre}$;_HBZ_ Tg, $n = 3$). The expression level of _Batf3_ in each type of mutant mouse was normalized to the expression level of _Gapdh_, and is shown as the relative ratio to that in WT mice. _Batf3_ is transcriptionally upregulated in splenic CD4$^+$ T cells from both _HBZ_ Tg and CARD11(E626K)$^{CD4\text{-}Cre}$;_HBZ_ Tg mice. **c** Western blotting for IRF4 in CD4$^+$ T cells. The protein level of IRF4 in each mouse type is normalized to the protein level of tubulin, and is shown as the relative ratio to that in WT mice. **d** Western blotting of cytoplasmic and nuclear fractions of IRF4 in CD4$^+$ T cells from each mouse type. Expression levels of IRF4 were normalized to those of tubulin in cytosolic fractions, and to those of TATA-binding protein (TBP) in nuclear fractions. The expression level in each type of mutant mouse is shown as the relative ratio to that in WT mice. **e** Summary of the relative levels of IRF4 in the nuclear fraction. A bar graph depicts the results from 3 replicate of Western blotting. _p_ values were calculated by the Tukey test after one-way ANOVA, and *** represent _p_ values less than 0.001, respectively.

pathway was selectively inhibited by _HBZ_ expression[34]. In line with this, the gene set associated with this pathway was negatively enriched in _HBZ_ Tg mice, but the finding was not statistically significant. The negative impact of _HBZ_ on the canonical NF-κB pathway might negate the enrichment of these gene sets in CARD11(E626K)$^{CD4\text{-}Cre}$;_HBZ_ Tg mice. In addition to the NF-κB pathway, cytokine signaling cascades were heavily enriched in CARD11(E626K)$^{CD4\text{-}Cre}$ mice. On the other hand, HBZ target gene sets were highly enriched in _HBZ_ Tg mice, but not in CARD11(E626K)$^{CD4\text{-}Cre}$ mice. In CARD11(E626K)$^{CD4\text{-}Cre}$;_HBZ_ Tg mice, CARD11 mutation and _HBZ_ expression cooperatively activated many signaling cascades, including those associated with gene sets related to the non-canonical NF-κB pathway, IRF4 targets, T-cell differentiation, cell proliferation, and adhesion or migration, which was consistent with these mice having the most severe phenotype.

We previously reported that mutations in genes encoding components of TCR-NF-κB signaling pathways were observed in 90% of samples from patients with acute-type ATL, and the median number of driver gene mutations in these samples was 10[16]. In addition, constitutive _HBZ_ expression was observed in acute-type ATL samples. We then compared the gene expressions in CARD11(E626K)$^{CD4\text{-}Cre}$;_HBZ_ Tg mice and acute-type ATL samples, and examined the combined effects of TCR-NF-κB signaling pathway activation and _HBZ_ expression on the pathogenesis of ATL. Many signaling cascades that were activated in human acute-type ATL samples were also activated in CARD11(E626K)$^{CD4\text{-}Cre}$;_HBZ_ Tg mice, indicating that TCR-NF-κB signaling pathway activation and _HBZ_ expression were together responsible for the basic molecular pathogenesis of acute-type ATL. In line with this, the BATF3/IRF4/HBZ transcriptional network[28] was enriched in both CARD11(E626K)$^{CD4\text{-}Cre}$;_HBZ_ Tg mice and human acute-type ATL samples. The proliferation of an ATL cell line was reported to be dependent on BATF3 and IRF4, which cooperatively drive ATL-specific gene expression, and HBZ was reported to induce BATF3 expression[28]. In CARD11(E626K)$^{CD4\text{-}Cre}$;_HBZ_ Tg mice, upregulation of BATF3 and IRF4 was observed in CD4$^+$ T cells. On the other hand, IRF4 was not upregulated in _HBZ_ Tg mice. CARD activation and _HBZ_ expression might be sufficient for the cooperative induction of proliferation/survival signals driven by BATF3 and IRF4 in ATL.

Patients with acute-type ATL often exhibit an extremely elevated lymphocyte count in PB, widespread lymphadenopathy, splenomegaly, and lymphocyte invasion of not only the lung but also the digestive tract, liver, adrenal grands, and brain. As this phenotype is much more severe than that in CARD11(E626K)$^{CD4\text{-}Cre}$;_HBZ_ Tg mice, additional mutations in T cells that already harbor NF-κB activation mutations and _HBZ_ expression might be required for the explosive increase in T cell number observed in acute-type ATL patients. In our previous analysis, _CARD11_ mutation was an early event in ATL development[16], suggesting that later mutations occur in cells that already harbor _CARD11_ mutations, thereby resulting in the full phenotype of aggressive

ATL. The full development of acute-type ATL might be partially due to the activation of signaling cascade genes, for example the NOTCH signaling cascade gene set, that was activated in human acute-type ATL samples but not in CARD11(E626K)$^{CD4\text{-}Cre}$;_HBZ_ Tg mice.

## Methods

**Mouse models.** Animal studies were approved and performed in accordance with the guidelines of University of Miyazaki Ethics Committee (#582). Targeting vector used to generate CARD11 mutant mice was constructed by recombination of the _mRosa26_ BAC (RP23-184A7) with cDNA containing the _Card11_E626K mutation, the homologue of _CARD11_(E626K), preceded by a _loxP_-flanked (FL) stop sequence (Supplementary Fig. 1a). The resulting vector was electroporated into C57BL/6 J embryonic stem (ES) cells. The recombinant ES cells were injected into ICR blastocysts, and subsequent chimera breeding resulted in CARD11(E626K)$^{stopFL}$ mice. Crossing these mice with _CD4_-Cre Tg mice[35] resulted in CARD11(E626K)$^{CD4\text{-}Cre}$ mice. To generate _HBZ_ Tg mice, _HBZ_ cDNA was cloned from the ATL cell line Su9T[36]. The clones were inserted into a cassette vector containing a mouse _CD4_ promoter/enhancer (Supplementary Fig. 1b)[37]. Electroporation and subsequent chimera breeding resulted in _HBZ_ Tg. Crossing CARD11(E626K)$^{CD4\text{-}Cre}$ and _HBZ_ Tg mice resulted in CARD11(E626K)$^{CD4\text{-}Cre}$;_HBZ_ Tg mice. These strains were backcrossed at least eight times onto the C57BL/6 background for this study. Male and female mice between 3 months and 12 months after birth were used in the experiments. An experimental flowchart is shown in Supplementary Fig. 16. The materials used in the experiments are shown in Supplementary Table 2.

**Gene expression profiling of mouse samples.** Splenic CD4$^+$CD44$^+$CD62L$^-$ Tem ($3 \times 10^4$ cells per mouse) were sorted from four types of mice: wild type (WT) ($n = 3$), CARD11(E626K)$^{CD4\text{-}Cre}$ ($n = 3$), _HBZ_ Tg ($n = 4$), and CARD11(E626K)$^{CD4\text{-}Cre}$;_HBZ_ Tg ($n = 3$). Total RNA extraction, cDNA library preparation, and sequencing were performed as described in the Supplementary Methods. Gene set enrichment analysis (GSEA)[22,23,38] was performed using a false discovery rate (FDR) cut-off of ≤0.25. Differentially expressed genes between sample groups were identified by Integrated Differential Expression and Pathway Analysis (iDEP) Tools, using cut-offs of fold change (FC) > 1.2 and an FDR of <0.1[39]. Functional annotation of identified genes was carried out in Enrichr[40].

**Comparison of gene expression signatures enriched in mutant mice and acute-type ATL samples.** To assess the extent to which the mouse models mimicked the molecular pathogenesis of ATL, we compared mouse and human GSEA results for the KEGG and HALLMARK gene sets, using their normalized enrichment score (NES) and FDR values. Human gene expression data were obtained from the healthy donors ($n = 3$), acute-type ATL patients ($n = 21$), and HTLV-1 carrier ($n = 3$) included in our previous report (EGAS00001001296)[16]. Using these expression data, GSEA of human samples was performed with the same protocol used with mouse samples, including the same gene sets and the same FDR cutoff value. To interpret the single and compound effects of the CARD mutant and _HBZ_ expression on ATL pathogenesis, each gene set in the mouse and human GSEA results was linked by name, plotted according to the mouse and human NES values, and highlighted according to the mouse and human FDR values. This was done for each combination of mouse genotype (WT/CARD11(E626K)$^{CD4\text{-}Cre}$/_HBZ_ Tg/CARD11(E626K)$^{CD4\text{-}Cre}$;_HBZ_ Tg) and human condition (healthy/acute-type ATL/HTLV-1 carrier).

**Statistics and reproducibility.** Differences between groups in terms of numbers and frequencies of cells were analyzed by ANOVA complemented with the post-hoc Tukey–Kramer test. Differences in animal survival, assessed by the Kaplan–Meier method, were analyzed by the log-rank test. Differences between frequencies of lymphadenopathy were assessed by Fisher's exact test with Benjamini–Hochberg correction. In vitro experiments were repeated three times individually and statistics of the measurements were calculated; in the case of only

two replications were made, the measurements were directly displayed. $p < 0.05$ indicated statistical significance. Statistical analysis was performed using R v.4.0.3.

**Reporting summary**. Further information on research design is available in the Nature Portfolio Reporting Summary linked to this article.

## Data availability

The cDNA sequences used are listed in Supplementary Table 2. The newly generated plasmid has been deposited in the DNA Data Bank of Japan (LC739268). Gene expression datasets obtained from human samples have been deposited at the European Genome-phenome Archive (EGAD00001001411). Gene expression datasets obtained from mouse samples have been deposited in the DNA Data Bank of Japan (DRA015050). Raw data for graphs are uploaded as Supplementary Data. Uncropped blots are provided as Supplementary Figs. 17–19. For data-sharing requests, please contact Kazuya Shimoda (kshimoda@med.miyazaki-u.ac.jp).

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

## Acknowledgements

The authors would like to thank M. Takeyama and Y. Nakamura (Axcelead Drug Discovery Partners, Inc.) for their technical assistance with the generation of CAR-D11(E626K)stopFL mice, and S. Saito, M. Matsushita, and T. Shinmori for their technical assistance with the experiments. This work was supported by Grants-in-Aid for Scientific Research (C) (17K09931, 20K08715) (T.K.) from the Japan Society for the Promotion of Science, by Research Grants (H28) (T.K.) from the Shinnihon Foundation of Advanced Medical Treatment Research, by Research Grants (R1) (T.K.) from The Japanese Society of Hematology, and by Research Grants (K. Shimoda; 20ck0106538h0001: S.O; 20ck0106409h0003) from the Japan Agency for Medical Research and Development (AMED). This study was supported by the Frontier Science Research Center, University of Miyazaki. Cell sorting experiments were supported by Y. Kawagoe.

## Author contributions

Conceptualization, T.K. and K. Shimoda; data curation, T.K., Y. Kogure, J.K., K.K., and T.O; methodology, T.K., K. Shide, A.K., Y.T., D.M., M. Sugiyama, J.K., and Y. Kogure; formal analysis, T.K., Y. Kogure, T.Y.-N., S.K., G.S., Y. Kitai, T.M. and A.Y.; investigation, T.K., K. Shide, A.K., Y.T., D.M., M. Sugiyama, K.A., K.M., M. Sekine, T.H., and Y. Kubuki; writing, T.K. and K. Shimoda; funding acquisition, T.K., K. Shimoda, and S.O.; supervision, K. Shimoda.

## Competing interests

K. Shimoda has received consulting fees from Novartis Pharma, Takeda Pharmaceutical, Bristol-Myers, Shire Japan, and Celgene, all outside the submitted work, and has received research grants from Perseus Proteomics, Pharma Essentia Japan KK, AbbVie GK,

Astellas Pharma, MSD, Chugai Pharmaceutical, Kyowa Kirin, Pfizer, Novartis Pharma, Otsuka Pharmaceutical, Asahi Kasei Medical, all outside the submitted work. K.K. holds stock in Asahi Genomics, has a patent for genetic alterations as a biomarker in T-cell lymphomas, and has received research funding from Chordia Therapeutics outside the submitted work. The remaining authors declare no competing interests.
