## [Peer Review File · Communications Biology]

Reviewers' comments:

Reviewer #1 (Remarks to the Author):

A deepened understanding of the signaling modalities promoting the development and onset of Adult T-cell leukemia/lymphoma (ATL) is needed to identify new therapy targets and strategies. In their work Kameda describe genetically engineered mouse models bearing t cell specific CARD11(E626K) and HBZ expression, respectively. They also generated double hit mice for CARD11(E626K) and HBZ and state that these genes cooperate to drive ATL development. The study mostly describes the effects of the genetic alterations on disease phenotype and delivers some indications for cooperative work of the two pathways using geneset enrichment analysis in comparison to human samples. The data and manuscript are adequately presented, yet lacking functional evidence. Altogether, I suggest the publication in CommsBio given some additional revision.

- 1) The authors state in line 165: "About half of the CARD11(E626K)CD4-Cre mice died at 6 or more months after birth (Figure 1A)". This would suggest that CARD11-Cre is less severe than double CARD11/HBZ-mut – However, the median survival according to figure 1 is 11.8 month. I suggest that rather than stating when "mice began to die" the median survival per group should be given in the main text.
- 2) The variance of survival of CARD11(E626K)CD4-Cre mice is high. The authors should comment why this is the case?
- 3) It is hard to follow how the mouse studies were setup. The article would gain by showing an experimental sketch of the mouse experiment- How many mice were injected. When were mice sacrificed? How many mice were left for survival analysis?
- 4) When describing the disease burden in mice, authors should introduce the reader (who might be more familiar with human than mouse anatomy) how to interpret the morphological changes in the mouse organs (e.g. spleen). How do the depicted features (number of splenic nucleated cells, spleen pulp architecture, etc.) vary from healthy mice and how can they be interpreted as a sign of lymphoproliferative disease.
- 5) It is in some parts of the manuscript hard to follow the authors rationales for the experiments performed. I recommend that the authors explain at the beginning of the respective section why the particular experiment/analysis was performed and also a short conclusion at the end of each section.
- 6) Why did the authors particularly analyse CD4+CD44+CD62L- Tem and CD4+CD25+ Treg cells? Why were numbers in the bone marrow and spleen summed up and not analysed separately? Was the composition in other T-cell compartments, e.g. CD8+ cells also altered?
- 7) The authors compare the number of CD44, FOXP3 and Ki67+ cells using histological sections. The authors should include a quantification/scoring and include an explanation of the quantification methodology.
- 8) For GSEA analysis, it would be interesting to see a comparison between double mutants and CARD11, HBZ-single mutant mice, respectively. These should be added into Figure 5.

Reviewer #2 (Remarks to the Author):

In this manuscript Takuro Kameda et al. Investigating the effect of single or combined expression of mutant CARD11 and HBZ in T cell lymphoproliferative diseases. Both of these genes were associated with lymphoproliferative diseases of adult T-cell leukemia/lymphoma (ATL) induced by HTLV. Transgenic mice generated expressing mutant CARD11 and HBZ or both were died much earlier than control mice associated with T cell proliferation and invasion in various organs,

especially lung. This effect was stronger in CARD11(E626K)CD4-Cre mice; HBZ Tg transgenic mice. RNAseq analysis combined with bioinformatic identified activation of NFkB and enrichment of proliferative genes in single and double transgenic mice. These results partly matching gene expression signature obtained from human ATL. Overall, the authors nicely demonstrated the importance of CARD11 and HBZ as well as combination of these two genes in ATL. Manuscript is well written and additional clarification may be needed for final publication.

Major comments:

Not all ATLs have CARD11 mutations in patients, any evidence that HBZ upregulates the expression of CARD11 in T cells? It is therefore possible that CARD11 is only a downstream target of HBZ involved in lymphoproliferation.

Since phenotype of transgenic mice is only partially resembles ATL patients, other mutations may be needed for stronger T cell proliferation in patients. Any evidence that CARD11 and HBZ mice altering genes associated with genomic stability. Cell transplantation then may be needed to further prove this observation.

As T cells mostly developed in thymus, any changes in size, structure or cellularity of this organ should be mentioned in this report.

While mice dies earlier than control groups in all transgenic groups, there is no indication of the cause of their early death. If this due to inflammatory response or lymphoproliferation, needs to be clarified and discussed.

As proliferation index is strongly enriched in all transgenic groups, is it possible to suggest that both CARD11 and HBZ acting as oncogenes to increase T cell pool in ATL cells upstream of cMYC?

Reviewer #3 (Remarks to the Author):

In this manuscript, Kameda et al. investigated the oncogenic ability of mutant CARD11 (E626K), one of most frequently mutated genes in adult T-cell leukemia (ATL), as well as its cooperative effect with HBZ expression using the CD4 promoter-driven HBZ transgenic mouse model. Remarkably, these mice developed T-cell lymphoma phenotypes such as lymphoproliferation and tissue invasion, which was accelerated in the double transgenic setting. The authors revealed that the activated genes enriched in double transgenic mice resembled that of acute ATL patients.

Overall, this study is nicely done. This study provides important insights which are relevant to the pathogenesis of human ATL. However, I think that their conclusion is still not clear. One main question is whether the phenotype observed in the HBZ/CARD11-double transgenic mice is distinct from the one in HBZ-single transgenic mice, or mutant CARD11 simply potentiates HBZ-induced tumorigenesis. Also, current study still lacks direct mechanistic evidences. Please see below for my general and specific comments.

General comments:

1. Difference between single and double transgenic settings

My main question is whether mutant CARD11 and HBZ have distinct oncogenic mechanisms, or CARD11 enhances the activation of the same oncogenic pathway induced by HBZ. For example, survival curve analysis (Figure 1) suggests that HBZ expression alone already shows high disease penetrance although tumor onset is slower than that in double transgenic setting. Similarly, although the percentage of lymphadenopathy in HBZ-single transgenic mice is lower than in the double transgenic mice at 6 months, it becomes comparable at 12 months (Figure 2). Additionally, the number of CD4+ cells and Tem cells in HBZ-single transgenic mice at later time-point (12 months) reached the same level or even higher than the level observed in the double transgenic setting at 6 months (Figure 4). Hence, the main difference between HBZ-single transgenic and HBZ/CARD11-double transgenic is the timing to develop tumors, but phenotypes observed may be similar. Indeed, GSEA shows that the same pathways were enriched in both single and double transgenic settings, although NES values are different. This further suggests that the main oncogenic pathways involved are the same between HBZ-single transgenic and HBZ/CARD11-double transgenic mice settings but the timing or degree of activation is different. In that case, is it more accurate to say that mutant CARD11 potentiates the HBZ-induced tumorigenesis by activating the same pathway. How can the authors address this point? For example, can the authors compare the results between different time-points (i.e., HBZ/CARD11-double transgenic at

6 months vs HBZ-single transgenic at 12 months) for the same data (phenotype, gene expression) and discuss about the difference and similarities?

2. Level of HBZ expression

One possibility to explain the difference in tumor onset is the expression level of HBZ. Currently, the authors showed the expression of HBZ only in the single transgenic mice (Supplemental Figure 2). However, this is not sufficient. My main question is whether HBZ expression level is comparable or different between HBZ-single and HBZ/CARD11-double transgenic settings. Indeed, it is very common to see a deference of tumor onset, if the expression level of transgene is different. For example, if the mutant CARD11 has an ability to increase the CD4 promoter activity, it may result in the increase of HBZ expression level and may affect tumor onset. In this case, it would be misleading to say that mutant CARD11 functionally synergizes with HBZ. Thus, it is necessary to confirm and exclude this possibility before they make a conclusion.

As additional question, has it been reported that tumor onset is different among different founder lines of HBZ transgenic mice? If there is a significant difference, it suggests that the dosage of HBZ may affect the onset in this model.

3. Lack of direct evidences

I am very cautious if the authors make a conclusion only based on correlation analysis such as GSEA, as this is still indirect evidence and may not be causal relationship. Can the authors show more direct evidences? For example, if they think that NF- κ B pathways are differentially activated in the single and double transgenic settings, can they analyze the status of canonical and non-canonical NF- κ B pathways by conventional methods (e.g., Western blot for NF- κ B signaling molecules, individual qRT-PCR for downstream targets, etc) and compare this result side-by-side across 4 settings (WT, CARD11, HBZ, and CARD11/HBZ)?

4. Genes and pathways that are uniquely regulated

It would be of interest to find if there are any genes or pathways that are uniquely regulated in either single or double transgenic setting. What genes/pathways were specifically up or downregulated in each setting? This point is currently not examined.

Specific comments:

5. Cellular phenotype

Is there any difference in T-cell morphology between single and double transgenic setting?

6. Supplemental Figure 2A

In the text, the authors mentioned about the increment of phosphorylation of CARD11 and p65 as well as cleaved BCL-10 level. However, the expression levels of the total proteins are quite different between WT and CARD11 mutation. Can they quantify the level and show the ratio of phosphorylation to total protein? Also, please include both nuclear and cytoplasmic markers in each panel to ensure that there is no significant contamination.

7. Peripheral blood cell count

Can the authors also show the number of peripheral blood cells, including CD4+ T cells, for each setting? It would be ideal if they can include multiple time-points.

8. Pathological analysis: Figure 1C, 2 and 3

It is unclear whether the spleen size was affected in each setting. Can they include scale bars? Also, I was aware that the authors used different cross-sections for each marker. But, I think that it is more appropriate to use the same sections to show that the same cells express multiple markers.

9. Infiltration of tumors cells in pulmonary tissues.

It is interesting that tumor cells in CARD11 transgenic mice were found in the lung. Any speculation on this?

10. Supplementary Figure 4

It seems that not all cells were stained with Ki-67. Maybe, can the authors show the ratio of Ki-67-positive cells per CD44 or Foxp3-positive cells? Is there any difference among single and double transgenic settings?

11. T-cell development

Do these transgenic mice affect the fate of CD4+CD8+ double-positive cells in thymus? How is the proportion of CD8+ T cells?

12. GSEA

Can the authors show the expression of genes included in each gene set by heatmap?

Reviewer #1 (Remarks to the Author):

A deepened understanding of the signaling modalities promoting the development and onset of Adult T-cell leukemia/lymphoma (ATL) is needed to identify new therapy targets and strategies. In their work Kameda describe genetically engineered mouse models bearing t cell specific CARD11(E626K) and HBZ expression, respectively. They also generated double hit mice for CARD11(E626K) and HBZ and state that these genes cooperate to drive ATL development. The study mostly describes the effects of the genetic alterations on disease phenotype and delivers some indications for cooperative work of the two pathways using geneset enrichment analysis in comparison to human samples. The data and manuscript are adequately presented, yet lacking functional evidence. Altogether, I suggest the publication in CommsBio given some additional revision.

Response: We would like to thank the reviewer for their positive and encouraging comments.

1) The authors state in line 165: "About half of the CARD11(E626K)CD4-Cre mice died at 6 or more months after birth (Figure 1A)". This would suggest that CARD11-Cre is less severe than double HBZ/CARD11-mut – However, the median survival according to figure 1 is 11.8 month. I suggest that rather than stating when "mice began to die" the median survival per group should be given in the main text.

Response: We completely agree with your opinion. In the revised manuscript we stated that the median survival times were 14, 8, and 6 months in CARD11 mutant mice, *HBZ* single transgenic mice, and *HBZ/CARD11* double transgenic mice, respectively.

2) The variance of survival of CARD11(E626K)CD4-Cre mice is high. The authors should comment why this is the case?

Response: The original OS curve data was derived from mice that were also subjected to blood cell count and pathological analysis. In revised Figure 1A we show the OS data from another mice cohort that we use only for survival observation and no other experiments. In this cohort, the variance of survival of each group mice is low.

3) It is hard to follow how the mouse studies were setup. The article would gain by showing an experimental sketch of the mouse experiment- How many mice were injected. When were mice sacrificed? How many mice were left for survival analysis?

Response: Thank you for your comments. We added an experimental flowchart as revised Supplemental figure 2. For the overall survival analysis, 31–88 mice in each group were observed. For blood cell count and cell surface marker analysis in BM, spleen, and LNs at 6 and 12 months, 13–21 and 5–12 mice in each mouse type were used, respectively. For pathological analysis at 6 and 12 months, 5 and 5 mice in each group, respectively, were used. The frequency of lymphadenopathy at 6 and 12 months was evaluated in 18–27 and 12–18 mice, respectively. RNA sequence analysis was performed in 3–4 mice from each

group.

4) When describing the disease burden in mice, authors should introduce the reader (who might be more familiar with human than mouse anatomy) how to interpret the morphological changes in the mouse organs (e.g. spleen). How do the depicted features (number of splenic nucleated cells, spleen pulp architecture, etc.) vary from healthy mice and how can they be interpreted as a sign of lymphoproliferative disease.

Response: Thank you for your advice. In response, we first described the microscopic features of each organ in WT mice, and compared them with those in each of the other mouse groups. We also transplanted splenic CD4⁺ cells of each mouse type into immunocompromised NOG mice to determine the visual signs of lymphoproliferative disease (Supplemental figure 8). Because small LNs contained mostly CD3⁺ cells, and CD3⁺ cells were detected in the spleens of recipient mice transplanted with *HBZ/CARD11* double transgenic CD4⁺ cells, we thought that lymphoproliferative neoplasms were the cause of death in *HBZ/CARD11* double transgenic mice. In contrast to *HBZ/CARD11* double transgenic mice, recipient mice transplanted with *CARD11* mutant-splenic CD4⁺ cells or *HBZ* single transgenic mouse CD4⁺ cells did not show a predominance of CD3⁺ cells in the spleen. As a result, we concluded that *CARD11* mutant mice and *HBZ* single transgenic mice developed severe inflammation rather than neoplasia.

5) It is a some parts of the manuscript hard to follow the authors rationales for the experiments performed. I recommend that the authors explain at the beginning of the respective section why the particular experiment/analysis was performed and also a short conclusion at the end of each section.

Response: Thank you for your comments. We stated the rationale for the experiments at the top of each section.

6) Why did the authors particularly analyse CD4⁺CD44⁺CD62L⁻ Tem and CD4⁺CD25⁺ Treg cells? Why were numbers in the bone marrow and spleen summed up and not analysed separately? Was the composition in other T-cell compartments, e.g. CD8⁺ cells also altered?

Response: ATL is a peripheral T-cell neoplasm characterized by the oncogenic proliferation of CD4⁺ T cells infected with HTLV-1. We focused on Tem and Treg cells in this manuscript, because HTLV-1 is usually detected in CD4⁺ T cells, and CD4⁺ T cells infected with HTLV-1 were positive for CCR4, CADM1, and CD25, but negative for CD45RA, indicating the effector memory T-cell phenotype. In addition, ATL cells express FOXP3, the glucocorticoid-induced TNF receptor (GITR), and the cytotoxic T-lymphocyte associated molecule-4 (CTLA-4), which are characteristic of regulatory T cells.

We presented the increment of Tem and Treg per mouse in *HBZ/CARD11* double

transgenic mice in our manuscript since these cells are distributed throughout many organs. According to your suggestions, we present the total numbers of Tregs and Tregs in each organ (BM and spleen) as Supplemental figure 5, together with the number of CD8⁺ cells (Figure 4). Compared to WT mice, there were fewer total CD8⁺ cells per mouse in HBZ single transgenic mice but similar numbers in CARD11 mutant mice and *HBZ/CARD11* double transgenic mice.

7) The authors compare the number of CD44, FOXP3 and Ki67+ cells using histological sections. The authors should include a quantification/scoring and include an explanation of the quantification methodology.

Response: We scored the numbers of CD44⁺, FOXP3⁺, and Ki67⁺ cells, together with the total nucleated cell numbers, in 5 microscopic views from each group of mice, and presented the percentages of CD44⁺, FOXP3⁺, and Ki67⁺ cells relative to the total number of nucleated cells in the LNs and lungs of each mouse group. We added these data to revised Supplemental figure 7 C and D. In LNs, the proportion of Tregs was increased in CARD11 mutant mice, and that of Tregs was increased in HBZ-single transgenic mice compared to WT mice. In *HBZ/CARD11* double transgenic mice, the proportion of Tregs, Tregs, and Ki-67⁺ cells was increased. In the perivascular interstitium of the lung, the proportions of Tregs, Tregs, and Ki-67⁺ cells in *HBZ/CARD11* double transgenic mice were increased, whereas they were comparable between WT mice and both CARD11 mutant and *HBZ* single transgenic mice.

8) For GSEA analysis, it would be interesting to see a comparison between double mutants and CARD11, HBZ-single mutant mice, respectively. These should be added into Figure 5.

Response: We added the GSEA analysis results of NF-KB pathway gene sets in *HBZ/CARD11* double transgenic mice compared with CARD11 mutant mice and *HBZ* single transgenic mice (revised Supplemental figure 10).

Reviewer #2 (Remarks to the Author):

In this manuscript Takuro Kameda et al. Investigating the effect of single or combined expression of mutant CARD11 and HBZ in T cell lymphoproliferative diseases. Both of these genes were associated with lymphoproliferative diseases of adult T-cell leukemia/lymphoma (ATL) induced by HTLV. Transgenic mice generated expressing mutant CARD11 and HBZ or both were died much earlier than control mice associated with T cell proliferation and invasion in various organs, especially lung. This effect was stronger in CARD11(E626K)CD4-Cre⁶²;HBZ Tg transgenic mice. RNAseq analysis combined with bioinformatic identified activation of NFkB and enrichment of proliferative genes in single and double transgenic mice. These results partly matching gene expression signature obtained from human ATL. Overall, the authors nicely demonstrated the importance of CARD11 and HBZ as well as combination of these two genes in ATL. Manuscript is well written and additional clarification may be needed for final publication.

Response: We would like to thank the reviewer for their positive and encouraging comments.

Major comments:

Not all ATLS have CARD11 mutations in patients, any evidence that HBZ upregulates the expression of CARD11 in T cells? It is therefore possible that CARD11 is only a downstream target of HBZ involved in lymphoproliferation.

Response: We would like to thank the reviewer for this valuable comment. To address the reviewer's concern, we compared the expression level of CARD11 between 4 strains of mice. The expression level in *HBZ*-single transgenic mice was comparable to that in WT mice, and that in *HBZ/CARD11*-double transgenic mice was comparable to that in *CARD11* mutant mice, indicating that *HBZ* expression had little effect on *CARD11* protein expression (revised Supplemental figure 3B).

Since phenotype of transgenic mice is only partially resembles ATL patients, other mutations may be needed for stronger T cell proliferation in patients. Any evidence that CARD11 and HBZ mice altering genes associated with genomic stability. Cell transplantation then may be needed to further prove this observation.

Response: We completely agree with your insights. GSEA analysis showed that gene sets whose expression was correlated with chromosomal instability were enriched in *HBZ/CARD11* double transgenic mice (revised Supplemental figure 13), suggesting that the combination of *HBZ* expression and *CARD11* mutation alter genes associated with genomic stability and contribute to the development of ATL.

According to your suggestion, we transplanted splenic CD4⁺ T cells from WT mice, *CARD11* mutation mice, *HBZ* single transgenic mice, or *HBZ/CARD11* double transgenic mice into immunocompromised NOG mice (revised Supplemental figure 8). Eighteen weeks after transplantation, all recipient mice were viable and no overt tumors

were visible. In recipient mice transplanted with CD4⁺ T cells from *CARD11/HBZ* double transgenic mice, CD3⁺ T cells from *CARD11/HBZ* double transgenic mice were detected in the spleen, whereas CD3⁺ T cells were rare in the spleens of recipient NOG mice transplanted with CD4⁺ T cells from WT, *CARD11* mutant mice, or *HBZ* single transgenic mice. Some small LNs were also detected in recipient mice transplanted with CD4⁺ T cells from *CARD11/HBZ* double transgenic mice. They mostly contained CD3⁺ T cells, and about 20% of CD3⁺ T cells expressed the proliferation marker Ki-67, suggesting that *CARD11/HBZ* double transgenic mice developed lymphoproliferative disease. The fact that *CARD11* mutant mice and *HBZ* single transgenic mice lacked splenic CD3⁺ cells as well as CD3⁺ T cells in LNs suggests that these mice developed severe inflammation, but not neoplasia.

As T cells mostly developed in thymus, any changes in size, structure or cellularity of this organ should be mentioned in this report.

Response: We would like to thank the reviewer for this important comment. The thymus was smaller in *HBZ* single transgenic mice and *HBZ/CARD11* double transgenic mice compared with WT mice at 6 months after birth (revised Supplemental figure 4). Accordingly, the nucleated cell number in one lobe of the thymus was lower in *HBZ/CARD11* double transgenic mice compared with WT mice. The size and cellularity of the thymus was unchanged in *CARD11* mutant mice.

While mice dies earlier than control groups in all transgenic groups, there is no indication of the cause of their early death. If this due to inflammatory response or lymphoproliferation, needs to be clarified and discussed.

Response: We appreciate the reviewer's insights. In *HBZ/CARD11* double transgenic mice, Ki-67⁺ Tem and Treg invaded many organs, including the lungs, and the alveolar air space of the lungs was diminished by this lymphocytic invasion. This was the direct cause of death in these mice. As stated above, small LNs contained CD3⁺ cells, and CD3⁺ cells were detected in the spleens of recipient mice transplanted with *HBZ/CARD11* double transgenic CD4⁺ cells. We therefore thought that lymphoproliferative neoplasms were the cause of death in *HBZ/CARD11* double transgenic mice.

In *CARD11* mutant mice and *HBZ* single transgenic mice, the invasion of Tem and Treg into many organs, including the lungs, was the direct cause of death as in *HBZ/CARD11* double transgenic mice. Contrary to *HBZ/CARD11* double transgenic mice, however, CD3⁺ T cells were not predominant in the spleens of recipient mice transplanted with *CARD11* mutant-splenic CD4⁺ T cells or *HBZ* single transgenic mice CD4⁺ T cells. We therefore thought that *CARD11* mutant mice and *HBZ* single transgenic mice did not develop neoplasms, but instead developed severe inflammation. *HBZ* transgenic mice were

previously described by several groups, and the phenotype of our *HBZ* single transgenic mice resembled those in previous reports. These reports showed that some mice, but not all, developed lymphoma 18 months after birth, and the cause of death was reported to be neoplasms (Satou et al. PLoS Pathog. 2011). Lymphoma may take a long time to develop in *HBZ* transgenic mice. Our *HBZ* single transgenic mice died of T-cell invasion of the lungs, and this might be why lymphoma did not develop in these mice in our study.

As proliferation index is strongly enriched in all transgenic groups, is it possible to suggest that both CARD11 and HBZ acting as oncogenes to increase T cell pool in ATL cells upstream of cMYC?

Response: MYC target genes were enriched in *HBZ/CARD11* double transgenic mice but not in *CARD11* mutant mice or *HBZ* single transgenic mice (revised Figure 6F). Myc induces transcription of the *E2F1*, *E2F2*, and *E2F3* genes. E2F target genes were also enriched in *HBZ/CARD11* double transgenic mice, but not in *CARD11* mutant mice or *HBZ* single transgenic mice. These observations suggest that a synergistic effect of *CARD11* mutation and *HBZ* expression induces an oncogenic signal upstream of cMYC, which increases the T-cell pool in ATL cells.

Reviewer #3 (Remarks to the Author):

In this manuscript, Kameda et al. investigated the oncogenic ability of mutant CARD11 (E626K), one of most frequently mutated genes in adult T-cell leukemia (ATL), as well as its cooperative effect with HBZ expression using the CD4 promoter-driven HBZ transgenic mouse model. Remarkably, these mice developed T-cell lymphoma phenotypes such as lymphoproliferation and tissue invasion, which was accelerated in the double transgenic setting. The authors revealed that the activated genes enriched in double transgenic mice resembled that of acute ATL patients.

Overall, this study is nicely done. This study provides important insights which are relevant to the pathogenesis of human ATL. However, I think that their conclusion is still not clear. One main question is whether the phenotype observed in the HBZ/CARD11-double transgenic mice is distinct from the one in HBZ-single transgenic mice, or mutant CARD11 simply potentiates HBZ-induced tumorigenesis. Also, current study still lacks direct mechanistic evidences. Please see below for my general and specific comments.

Response: We would like to thank the reviewer for their positive and encouraging comments. As stated in the responses to each specific query below, the phenotype observed in *HBZ/CARD11* double transgenic mice was similar to that in *HBZ* single transgenic mice, and they shared common activated signaling cascades. On the other hand, there are uniquely activated signals in *HBZ/CARD11* double transgenic mice. For instance, the IRF4/BATF3/HBZ transcriptional network, which was reported to be essential for the proliferation of cells from an ATL cell line, was activated in *HBZ/CARD11* double transgenic mice but not in *HBZ* single transgenic mice. This is due to the increased amount of IRF4 in the former but not the latter. We also present western blot and qPCR data in the revised manuscript.

General comments:

1. Difference between single and double transgenic settings

My main question is whether mutant CARD11 and HBZ have distinct oncogenic mechanisms, or CARD11 enhances the activation of the same oncogenic pathway induced by HBZ. For example, survival curve analysis (Figure 1) suggests that HBZ expression alone already shows high disease penetrance although tumor onset is slower than that in double transgenic setting. Similarly, although the percentage of lymphadenopathy in HBZ-single transgenic mice is lower than in the double transgenic mice at 6 months, it becomes comparable at 12 months (Figure 2). Additionally, the number of CD4+ cells and Tem cells in HBZ-single transgenic mice at later time-point (12 months) reached the same level or even higher than the level observed in the double transgenic setting at 6 months (Figure 4). Hence, the main difference between HBZ-single transgenic and HBZ/CARD11-double transgenic is the timing to develop tumors, but phenotypes observed may be similar. Indeed, GSEA shows that the same pathways were enriched in both single and double transgenic settings, although NES values are different. This further suggests that the main oncogenic pathways involved are the same between HBZ-single transgenic and HBZ/CARD11-

double transgenic mice settings but the timing or degree of activation is different. In that case, is it more accurate to say that mutant CARD11 potentiates the HBZ-induced tumorigenesis by activating the same pathway. How can the authors address this point? For example, can the authors compare the results between different time-points (i.e., HBZ/CARD11-double transgenic at 6 months vs HBZ-single transgenic at 12 months) for the same data (phenotype, gene expression) and discuss about the difference and similarities?

Response: We would like to thank the reviewer for these comments that clearly helped improve the manuscript. Both *HBZ* single transgenic mice and *HBZ/CARD11* double transgenic mice exhibited similar phenotypes, and lymphocyte invasion into many organs was observed. As pointed out, the major phenotypic difference between them was disease severity or timing. At 6 months after birth, *HBZ* single transgenic mice exhibited a milder phenotype than *HBZ/CARD11* double transgenic mice, but the phenotypic severity was similar between *HBZ* single transgenic mice at 12 months after birth and *HBZ/CARD11* double transgenic mice at 6 months after birth. According to your suggestion, we compared gene expression in *HBZ* single transgenic mice at 8–12 months with that in *HBZ/CARD11* double transgenic mice at 4–6 months (Supplemental figure 14). A total of 430 genes and 227 genes were up- and downregulated, respectively, in both *HBZ* single transgenic mice at 8–12 months and *HBZ/CARD11* double transgenic mice at 4–6 months. Gene sets targeted by *HBZ*, which were thought to be the driving force behind oncogenesis in *HBZ* single transgenic mice, were highly enriched in both *HBZ* single transgenic mice at 8–12 months and in *HBZ/CARD11* double transgenic mice at 4–6 months. This suggests that both types of mice share the same oncogenic mechanism.

Even though many genes were up- or downregulated in both *HBZ* single-transgenic mice at 8–12 months and *HBZ/CARD11* double transgenic mice at 4–6 months, there were genes that were uniquely up- or downregulated in only one type. Additionally, PCA analysis demonstrated that *HBZ* single transgenic mice at 8–12 months comprised a different gene expression hierarchy branch than *HBZ/CARD11* double transgenic mice at 4–6 months. The uniquely upregulated gene sets in *HBZ/CARD11* double transgenic mice at 4–6 months (n=78) were more enriched in acute-type ATL patient samples than in those from HTLV-1 carriers. On the other hand, uniquely upregulated gene sets in *HBZ* single transgenic mice at 8–12 months (n=2970) did not exhibit this enrichment. These observations suggested that *HBZ/CARD11* double transgenic mice and *HBZ* single transgenic mice had some oncogenic signals in common, but there was not complete overlap. One of the uniquely activated signals in *HBZ/CARD11* double transgenic mice is related to the *BATF3/IRF4/HBZ* transcriptional network, which was reported to be essential for the proliferation of cells from an ATL cell line (Nakagawa et al., Cancer Cell

2018)(Figure 6C). The BATF3/IRF4/HBZ transcriptional network was not activated in CARD11 mutant mice or *HBZ* single transgenic mice. In humans, this network was enriched in acute-type ATL patient samples but not in those from HTLV-1 carriers.

BATF3 was upregulated in both *HBZ* single transgenic mice and *HBZ/CARD11* double transgenic mice (revised Figure 6D). There was a 2.0-fold increase in IRF-4 protein levels in CD4⁺ cells from *HBZ/CARD11* double transgenic mice compared with those from WT mice (revised Figure 6E). Compared to WT mice, an increase in nuclear levels of IRF4 was observed in *HBZ/CARD11* double transgenic mice but not in *HBZ* single transgenic mice. In line with this, many IRF4 target gene sets were enriched in *HBZ/CARD11* double transgenic mice, but not in *HBZ* single transgenic mice (Supplemental figure 15). In *HBZ/CARD11* double transgenic mice, the elevated expression of BATF3 and IRF4 might induce the activation of the BATF3/IRF4/HBZ transcriptional network. In *HBZ* single transgenic mice, on the other hand, the expression of BATF3 was upregulated but that of IRF4 was unchanged.

Taken together, the phenotypes of *HBZ* single transgenic mice and *HBZ/CARD11* double transgenic mice were similar, and many of the same signaling pathways were activated in both. In addition, a uniquely activated cascade induced by the combination of CARD11 mutation and HBZ expression, for example the BATF3/IRF4/HBZ transcriptional network, might be required for the development of ATL.

2. Level of HBZ expression

One possibility to explain the difference in tumor onset is the expression level of HBZ. Currently, the authors showed the expression of HBZ only in the single transgenic mice (Supplemental Figure 2). However, this is not sufficient. My main question is whether HBZ expression level is comparable or different between HBZ-single and HBZ/CARD11-double transgenic settings. Indeed, it is very common to see a deference of tumor onset, if the expression level of transgene is different. For example, if the mutant CARD11 has an ability to increase the CD4 promoter activity, it may result in the increase of HBZ expression level and may affect tumor onset. In this case, it would be misleading to say that mutant CARD11 functionally synergizes with HBZ. Thus, it is necessary to confirm and exclude this possibility before they make a conclusion.

As additional question, has it been reported that tumor onset is different among different founder lines of HBZ transgenic mice? If there is a significant difference, it suggests that the dosage of HBZ may affect the onset in this model.

Response: We would like to thank the reviewer for the important comment on addressing the cause of lymphoproliferative disease in *HBZ/CARD11* double transgenic mice. As shown in revised Supplemental figure 3B, the HBZ expression level was comparable between *HBZ* single transgenic mice and *HBZ/CARD11* double transgenic mice.

Satou and Matsuoka reported 3 lines of *HBZ* transgenic mice (PLoS Pathogens 2011). The #9 strain, which showed the highest HBZ expression among the 3 lines, had a shorter lifespan than the other strains; however, the #9 strain consisted of 7 mice, while the other 2 strains consisted of 3 and 4 mice, respectively. The difference in tumor onset between the different founder lines was not mentioned in their report.

3. Lack of direct evidences

*I am very cautious if the authors make a conclusion only based on correlation analysis such as GSEA, as this is still indirect evidence and may not be causal relationship. Can the authors show more direct evidences? For example, if they think that NF- κ B pathways are differentially activated in the single and double transgenic settings, can they analyze the status of canonical and non-canonical NF- κ B pathways by conventional methods (e.g., Western blot for NF- κ B signaling molecules, individual qRT-PCR for downstream targets, etc) and compare this result side-by-side across 4 settings (WT, *HBZ/CARD11*, and *HBZ/CARD11*)?*

Response: We present the results of the western blot analysis of RelA (p65) and RelB in the nuclear and cytosolic fractions of CD4⁺ splenic T cells in revised Supplemental figure 11. High nuclear levels of RelA (p65), but not RelB, were detected in splenic CD4⁺ cells from *CARD11* mutant mice, indicating that the canonical NF- κ B pathway was activated by *CARD11* mutation. In splenic CD4⁺ cells from *HBZ/CARD11* double transgenic mice, high nuclear levels of RelB suggested activation of the non-canonical NF- κ B pathway. We also added the results of western blot analysis of IRF4 and qPCR data for BATF3 in revised Figure 6.

4. Genes and pathways that are uniquely regulated

It would be of interest to find if there are any genes or pathways that are uniquely regulated in either single or double transgenic setting. What genes/pathways were specifically up or downregulated in each setting? This point is currently not examined.

Response: Compared with WT mice, the *CARD11* mutant, *HBZ* single transgenic, and *HBZ/CARD11* double transgenic mice showed upregulation of 17, 894, and 522 genes, respectively, and downregulation of 24, 513, and 305 genes, respectively (revised Figure

5A). KEGG pathway analysis showed that TGF-beta signaling, mTORC1 signaling, inflammatory response, and apical junction was only activated in CARD11 mutant mice, mitotic spindle was only activated in *HBZ* single transgenic mice, and PI3/AKT/mTOR signaling, interferon alpha response, Myc target, E2F target, and G2M checkpoint were only activated in *HBZ/CARD11* double transgenic mice (revised Figure 5B). In contrast, TNF alpha signaling via NF-kappa B, IL-2/STAT5 signaling, and interferon gamma response were activated in CARD11 mutant mice, *HBZ* single transgenic mice, and *HBZ/CARD11* double transgenic mice. The p53 pathway and IL-6/JAK/STAT3 signaling were activated in both *HBZ* single transgenic mice and *HBZ/CARD11* double transgenic mice, but not in CARD11 mutant mice.

Specific comments:

5. Cellular phenotype

Is there any difference in T-cell morphology between single and double transgenic setting?

Response: Flower-like cells and pleomorphic nuclear features are among the characteristics of ATL cells. Morphologically abnormal or atypical cells were never seen in WT mice, and were rarely observed in CARD11 mutant mice or *HBZ* single transgenic mice. In *HBZ/CARD11* double transgenic mice, about 5% of lymphocytes demonstrated pleomorphic nuclear features, whereas morphological characteristics of other lymphocytes were normal. We added these data to revised Figure 1C.

6. Supplemental Figure 2A

In the text, the authors mentioned about the increment of phosphorylation of CARD11 and p65 as well as cleaved BCL-10 level. However, the expression levels of the total proteins are quite different between WT and CARD11 mutation. Can they quantify the level and show the ratio of phosphorylation to total protein? Also, please include both nuclear and cytoplasmic markers in each panel to ensure that there is no significant contamination.

Response: Since mutant CARD11 is exogenously expressed, the total of WT and mutant CARD11 protein is much higher in CD4⁺ cells from CARD11 mutant mice compared with WT mice (revised Supplemental figure 3A). The phospho-CRRD11/total CARD11 protein ratio was increased by 1.2-fold in CARD11 mutant CD4⁺ cells relative to those from WT mice. We also added nuclear and cytoplasmic markers.

7. Peripheral blood cell count

Can the authors also show the number of peripheral blood cells, including CD4+ T cells, for each setting? It

would be ideal if they can include multiple time-points.

Response: At 3 months after birth, the peripheral blood leukocyte number was increased in CARD mutant mice and was decreased in *HBZ* single transgenic mice and *HBZ/CARD11* double transgenic mice compared to WT mice. Accordingly, the number of CD4⁺ cells was increased in CARD mutant mice and was decreased in *HBZ* single transgenic mice and *HBZ/CARD11* double transgenic mice compared to WT mice. There were no differences in Hb value or platelet count between the 4 types of mice. We presented the white blood cell counts and the number of CD4⁺ T cells in peripheral blood at 3 and 6 months after birth in revised Figure 1B.

8. Pathological analysis: Figure 1C, 2 and 3

It is unclear whether the spleen size was affected in each setting. Can they include scale bars? Also, I was aware that the authors used different cross-sections for each marker. But, I think that it is more appropriate to use the same sections to show that the same cells express multiple markers.

Response: We show pathological sections, including scale bars, in revised Figure 1E, 2B, and 3. We substituted several slides images in Figure 1E and Figure 3 to the consecutive sections. In Figure 2B, the original presented figures were consecutive sections showing immunohistochemistry analysis.

9. Infiltration of tumors cells in pulmonary tissues.

It is interesting that tumor cells in CARD11 transgenic mice were found in the lung. Any speculation on this?

Response: GSEA analysis showed that the expression of ICAM, EPCAM, ITGB6, and CCR2 was increased in CARD11 mutant CD4⁺ cells. Even though western blot analysis, immunohistostaining, and qPCR did not show that these molecules were upregulated in CARD11 mutant mice, the expression of these adhesion- and migration-related genes might induce the severe inflammation in CARD11 mutant mice.

10. Supplementary Figure 4

It seems that not all cells were stained with Ki-67. Maybe, can the authors show the ratio of Ki-67-positive cells per CD44 or Foxp3-positive cells? Is there any difference among single and double transgenic settings?

Response: We counted the ratio of Ki-67⁺ cells per CD44⁺ or FOXP3⁺ cells in consecutive LN sections (revised Supplemental figure 7D). In the CD44⁺ cell population, Ki-67⁺ cells were more often observed in *HBZ* single transgenic mice and *HBZ/CARD11* double transgenic mice, in which the ratio of Ki-67⁺ cells among Tregs was 19% and 27%,

respectively, compared with WT mice, in which the ratio was 4%. In the FOXP3⁺ cell population, Ki-67⁺ cells were more often detected in HBZ/CARD11 double transgenic mice. 41% of FOXP3⁺ cells were positive for Ki-67 in LNs from HBZ/CARD11 double transgenic mice, whereas the positivity ratio was 23% in LNs from CARD11 and HBZ single transgenic mice, and 2% in LNs of WT mice.

We were unable to obtain this ratio in consecutive lung sections, and we are sorry that we cannot present it.

11. T-cell development

Do these transgenic mice affect the fate of CD4⁺CD8⁺ double-positive cells in thymus? How is the proportion of CD8⁺ T cells?

Response: We would like to thank the reviewer for this important comment. The thymus was shrunken in *HBZ* single transgenic mice and *HBZ/CARD11* double transgenic mice compared with WT mice at 6 months after birth. Accordingly, the nucleated cell number in one lobe of the thymus was lower in *HBZ/CARD11* double transgenic mice compared with WT mice. The size and cellularity of the thymus was unchanged in *CARD11* mutant mice.

The number of CD4⁺8⁺ T cells and CD4⁺8⁺ T cells in one lobe of the thymus was comparable between each group mice. We presented these data as revised Supplemental figure 4.

12. GSEA

Can the authors show the expression of genes included in each gene set by heatmap?

Response: In the GSEA analysis, we now show the expression of genes in each set by a heatmap.

REVIEWERS' COMMENTS:

Reviewer #1 (Remarks to the Author):

The authors addressed most of my comments in an acceptable manner and I would accept the article for publication.

Yet, the manuscript Figures appear overloaded and hard to read. I highly recommend to revise this.

Reviewer #2 (Remarks to the Author):

The authors nicely addressed my comments and this manuscript is now acceptable from my side for publication.

Reviewer #3 (Remarks to the Author):

The authors have addressed all concerns.

Reviewer #1 (Remarks to the Author):

The authors addressed most of my comments in an acceptable manner and I would accept the article for publication.

Yet, the manuscript Figures appear overloaded and hard to read. I highly recommend to revise this.

Response : Thank you for your comments. We split figures with too much information into two figures to make them easier to read. In addition, some figures were moved to Supplementary Information. Specifically, we split the previous Figures 1, 5, and 6 into two figures each, and moved part of previous Figure 6 to Supplemental Information.